

# Determination of high resolution spatio-temporal glacier motion fields from time-lapse sequences

Ellen Schwalbe[1], Hans-Gerd Maas[1]

[1]Institute of Photogrammetry and Remote sensing, Technische Universität Dresden, Dresden, 01069, Germany

*Correspondence to*: Ellen Schwalbe (ellen.schwalbe@tu-dresden.de)

**Abstract.**

This paper presents a comprehensive method for the determination of motion vector fields of glaciers at high spatial and temporal resolution. These vector fields can be derived from monocular terrestrial camera image sequences and are a valuable data source for glaciological analysis of the motion behaviour of glaciers. The measurement concepts for the

acquisition of image sequences are presented, and an automated monoscopic image sequence processing chain is developed. Motion vector fields can be derived with high precision by applying automatic sub-pixel-accuracy image matching techniques on grey value patterns in the image sequences. Well-established matching techniques have been adapted to the special characteristics of the glacier data in order to achieve high reliability in automatic image sequence processing, including the handling of moving shadows as well as motion effects induced by small instabilities in the camera setup.

Suitable geo-referencing techniques were developed to transform image measurements into a reference coordinate system.

The result of the monoscopic image sequence analysis is a dense raster of glacier surface point trajectories for each image sequence. Each translation vector component in these trajectories can be determined with an accuracy of some centimetres for points at a distance of several kilometres from the camera. Extensive practical validation experiments show that motion

vector and trajectory fields derived from monocular image sequences can be used for the determination of high resolution velocity fields of glaciers, for the analysis of the effects of tides on glacier movement, for the investigation of a glacier's motion behaviour during calving events, for the determination of the position and migration of the grounding line and for the detection of sub glacial channels during glacier lake outburst floods.

## 1 Introduction

For almost a century, terrestrial photogrammetry has been an important measurement method for glaciology research. Early basics for the use of photogrammetric methods for velocity measurements of glaciers were provided by Finsterwalder (1931) and Pillewizer (1938). For glaciers in the Pamir, the Himalaya, Norway and the Alps, velocity profiles were derived by repeated measurements. With the advance of the technical development of the cameras, the acquisition intervals for the photogrammetric measurements were getting shorter. For the investigation of short-term motion behaviour of glaciers,



automatically recording analogue cameras were used from the 1970s onwards. The application potential of these cameras for the measurement of glaciers is described in Flotron (1973). Further examples for the use of automatic cameras are the photogrammetric time-lapse measurements on Columbia Glacier (Alaska) by Krimmel & Rasmussen (1986), which were conducted with a temporal resolution of three records per day, and the investigations of the Variegated-Glacier (Alaska) by
Harrison et al. (1986).

With the appearance of digital cameras and the rapid development of their sensors, photogrammetric tools have been available since the end of the $20^{th}$ century, which allow for an automatic acquisition of data sets with not only high spatial but also high temporal resolution. Because of further technical improvements of digital cameras and developments in
automatic image matching techniques, terrestrial image sequence analysis became a valuable tool for the investigation of glaciers during the last 20 years. Phenomena with short-term movement variations requiring a temporal resolution of hours (tidal influence on the glacier movement) or even minutes (calving events) can now be investigated efficiently. Early applications of photogrammetry-based methods for the determination of glacier velocities from terrestrial digital time-lapse measurements have been published by Maas et al. (2006) and Ahn & Box (2010). Since then, time lapse- measurements
were increasingly used for glacier motion analysis (examples are: Eiken and Sund, 2012; Rivera et. al., 2012; Danielson and Sharp, 2013; James et al., 2014; James et al., 2016).

Beyond glaciology applications, time-lapse imagery is also of great interest for geomorphologic investigations. Parajka et al. (2012) e.g. used time-lapse images recorded with an hourly time interval to observe the snow cover in small catchments.
Matsuoka et al. (2014) conducted a five year investigation with time-lapse images to observe soil movement due to frost creep and heave at alpine slopes, taking images every four hours. To monitor and analyse large wood loads in rivers, Kramer & Wohl (2014) recorded images every minute and manually analysed them regarding the presence or absence of wood. Nichols et al. (2016) used time-lapse images to qualify gully erosion. They identified the subsurface erosion as a cause for gully headwall retreat. Thereby the time-lapse measurements were considered as an important supplement besides
topographic survey with total station and Lidar.

The simple use of time-lapse cameras as a visual observation tool may already be a great help for environmental investigations. However, beyond that, they have the potential to also deliver precise measurements with high temporal and spatial resolution when applying appropriate photogrammetric image sequence processing techniques. In this paper we
introduce a comprehensive method for the determination of motion vector fields from terrestrial time-lapse image sequences. The method is designed for the observation of glaciers, but might also be adapted to other environmental motion analysis tasks.



The main part of the paper will be about methodological issues of photogrammetric image sequence analyses (section 2). After introducing the basic concept of the method (section 2.1), several aspects regarding the measurement setup for time-lapse image acquisition will be discussed (section 2.2). The analysis of the image sequences consists of two main parts. The first one comprises the determination of motion vectors from the images (section 2.3) and the second one the scaling and

geo-referencing of the measurement values (section 2.4). The methodological section will be completed with a discussion of the accuracy potential of the presented method (section 2.5). In the second major part of the paper we present the results of different glaciology research pilot studies that were conducted on the basis of glacier motion data obtained by applying the presented time-lapse measurement method and image sequence analysis approach (section 3).

## 2 Methodological Approach

In the following section the monoscopic image sequence analysis will be described in detail. The approach consists of two main tasks: The first one comprises image based measurements (also referred to as 'measurements in image space'), and the second one the scaling and geo-referencing of these measurements (also referred to as 'translation into object space'). Furthermore the main aspects of accuracy in monoscopic image sequence analysis will be discussed. We limit ourselves to monoscopic image motion capture and processing delivering two-dimensional velocity field information here, as the

glaciology phenomena observed in the practical experiments do not show significant across-track motion and can thus be well described in 2D. The method can be extended to 3D trajectory measurements from stereoscopic image sequences straightforwardly by inserting an additional stereo image matching step into the data processing chain.

### 2.1 Monoscopic image sequence approach - basic concept

In the monoscopic time-lapse measurements, a single firmly installed camera observes the area of interest and continuously

records images at a pre-selected time interval. The obtained image sequence is the basis for the determination of glacier motion. Applying appropriate feature tracking algorithms, the position of a large number of moving glacier surface points can be obtained for each image of the sequence. Thus, the glacier point's motion vectors in image space can be derived for each individual pair of sequence images. A trajectory of a glacier surface feature is obtained by tracking points through multiple images of the sequence.

The motion vectors or trajectories measured in image space have to be transferred into object space. This means they need to be scaled, and the 3D position of the trajectory origins need to be determined. In a stereo approach, a 3D position is determined for each homologous point pair detected in the stereo images (Figure 1). This 3D position also delivers the distance of the point to the cameras, which is required for scaling the measurements from pixels to metric values. However,

stereo image measurements on rugged glacier surfaces will often suffer from significant de-correlation, limiting the precision and reliability of stereo image sequence processing. Therefore, monoscopic image sequences processing is more appropriate





for phenomena, which are mainly two-dimensional. For scaling and geo-referencing of monoscopic image sequence measurements an approximate digital surface model (DSM) of the glacier is required as well as the camera position and orientation parameters. Based on this information, for each pixel an image ray can be reconstructed and intersected with the DSM (Figure 1). This delivers the distance between glacier and camera for the scaling as well as the 3D glacier point

position for the geo-referencing of the measurement value. With knowledge on the time interval between consecutive images, metric velocity values can be calculated from the motion vectors or trajectories.

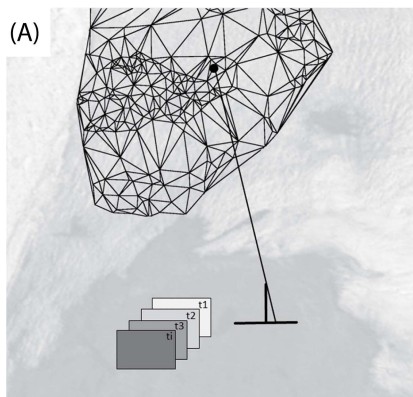 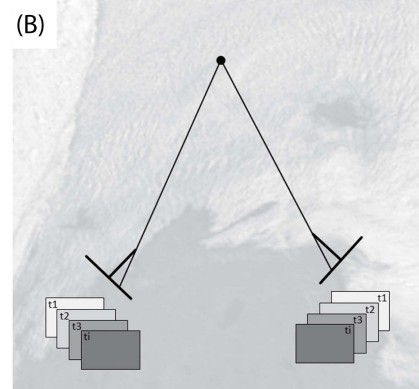

**Figure 1: Monoscopic (A) vs. stereoscopic (B) image sequence analysis.**

## 2.2 Measurement Setup

Figure 2 shows the basic measurement setup (A) and the required visual content of the measurement image acquired by the time-lapse camera (B) to meet the demands of successful precise data processing. The main requirement for a camera to be used for glacier motion analysis is the ability to autonomously record images at defined time intervals over a certain period

of time. It should therefore be equipped with a programmable timer as well as a weatherproof housing, sufficient power supply (e.g. a batteries supported by a solar panel) and sufficient data storage capacity.

For a flexible and stable installation of time-lapse cameras, tripods thoroughly covered with stones have proven to be suitable. To determine both horizontal and vertical components of glacier motion and to avoid occlusions on the glacier

surface, an elevated camera position is required, which allows for an oblique viewing angle on the glacier surface. Despite the stable installation of a time-lapse camera, slight camera motion (mainly induced by wind and temperature effects) cannot be completely prevented. To compensate for this, it is necessary to have static points visible in the images, which can be used for correcting the influence of camera movement (see section 2.3.3). These may be natural features in the image fore- and/or background, but in most cases artificial marks placed in the foreground (Figure 2, B) turned out to be more accurate

and reliable.





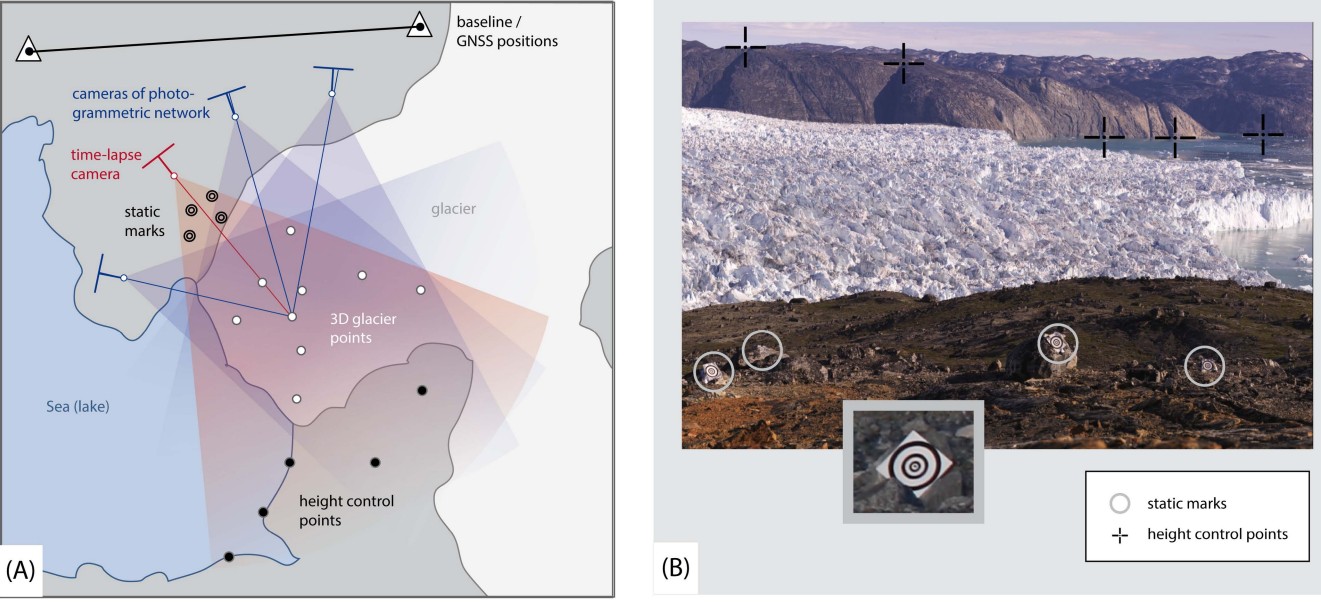

**Figure 2: Scheme of measurement setup (A), measurement image of time lapse camera (B)**

For the geo-referencing and scaling of measurement values of the monoscopic time-lapse camera approach, the geometric relation between image and glacier surface must be known. Thus, a digital surface model of the glacier and the orientation parameters of the time-lapse camera have to be determined. This can be achieved by establishing a local photogrammetric network (consisting of several convergent images taken from different positions as shown in Figure 2) plus some additional

10 measurements. Together with one time-lapse camera image, these form a photogrammetric network, which allows for the simultaneous determination of glacier surface point coordinates and camera orientation parameters by photogrammetric bundle adjustment (see e.g. Kraus, 2007 or Luhmann et.al., 2006). A minimum constellation to transform the results into a scaled and horizontal world coordinate system (i.e. to establish a geodetic datum) is given by the knowledge of two camera positions (preferably those forming the longest baseline) and one height control point (preferably at a far distance). However,

it is recommendable to use redundant information in the process of geo-referencing. Therein, different variants of measurement setups are possible, depending on the local terrain topography:

A minimal geo-referencing measurement equipment consists of a hand held GNSS (global navigation satellite system) device and a hand held laser distance measuring device. This instrumentation is sufficient, if a high accurate elevation

reference is not required for a certain measurement task. Using the hand held GNSS, at least two camera positions (preferably those, which form the longest baseline) and at least one control point (ideally 5-10 for redundancy) in the



background of the measurement object need to be measured. With the hand held laser distance measuring device precise distances between camera positions can be determined. The distance between the camera positions (called base length) defines the scale for translating image measurements into object space, and errors in the base length will propagate linearly into the length of determined motion vectors and trajectories. A special, advantageous case occurs when the shoreline of a

lake or the sea is visible in the time-lapse images. The control point measurements can then be reduced to a single water level measurement that delivers the Z-coordinate for several height control points along the shore line. These height control points form a good basis for defining the world coordinate system horizon.

If a higher positioning accuracy is required, the camera position determination via hand held GNSS measurements should be

replaced by differential GNSS measurements. Besides an accurate elevation reference, camera positions measured with an accuracy of a few centimeters ensure a more accurate scaling of the photogrammetric network, thus making additional laser distance scale measurements dispensable. Depending on the cameras resolution, it might still be sufficient to measure control points using hand held GNSS if they are located in adequate distance from the time-lapse camera.

The highest measurement effort is required for measuring environments in which the control points are inaccessible or very distant. In these cases control points need to be determined via triangulation. For this purpose a base line has to be established whose coordinates are measured via GNSS. From the base line positions, angle measurements are conducted by a tachymeter instrument to determine coordinates of control points via spatial intersection. Camera positions can then be transformed from a local horizontal coordinate system into a global coordinate system by the GNSS measurements.

### 2.3 Image Sequence analysis

This section will focus on the measurements in image space. The basis is a recorded image sequence of a moving glacier. To derive motion vectors from an image sequence, an appropriate feature tracking algorithm is required as well as an appropriate tracking strategy. To obtain reliable results, suitable methods need to be developed to compensate for the main

error influences, including the effects of camera motion as well as effects of the motion of shadows on the glacier.

### 2.3.1 Feature tracking

For glacier surface point tracking a wide range of algorithms is available (see e.g. Brown, 1992 and Zitová & Flusser, 2003). Generally they can be separated into feature based and area based methods. Area based methods establishing

correspondences between small image patches by minimizing some cost functions are able to provide results with sub-pixel accuracy and allow for a dense and regular distribution of measurement points over the image area of interest. Feature based techniques extract discrete features from the image and track those by comparing values of a defined feature vector. We



chose an area based method for the task of glacier motion determination due to the higher accuracy potential. The chosen method depicts a combination of cross correlation (e.g. Lewis, 1995) and least squares matching (LSM) (Förstner, 1982; Ackermann, 1984; Grün, 1985). The cross correlation method is a fast option to obtain approximate values that are required for LSM. The advantage of LSM is that it is an adaptive, robust and accurate approach. It directly estimates sub-pixel

accuracy patch translation parameters and simultaneously delivers figures on their accuracy and reliability. Furthermore, LSM is also capable to consider rotations and linear distortions of patches.

In order to determine glacier motion fields, a dense raster of points needs to be defined in the first image of the image sequence. For each of these measurement points a patch has to be defined as well as a search area before applying the actual

tracking algorithm. The patch contains the area of pixels around the measurement point to be tracked, and the search area defines the region were the corresponding point might be located in the subsequent image. There are special demands on both of them when monitoring glaciers:

The resulting motion vector for a certain patch will be an interpolated value for the area on the glacier surface that is covered

by the patch. When terrestrially observing glaciers the view to the glacier surface will usually be rather oblique. Thus a quadratic patch that contains an adequate amount of pixels will cover an area on the glacier surface that has a large extension in depth direction (i.e. across glacier motion). This is especially critical for remote measurement points. In order to address this problem, a specific rectangular shape is chosen for each patch (Figure 3). To automatically define the optimal patch size, two parameters are predefined. The fist one is a number of pixels a search patch should contain to ensure stable matching,

and the second one is a maximal difference in depth a patch should cover to avoid interpolation over large areas. While the depth parameter restricts the extension of the patch in y-direction, the given number of pixels lets the patch proportionately grow in x-direction. The DSM of the glacier surface that is required to obtain the necessary distance information for patch shape restriction can be provided as described in section 2.4.2.



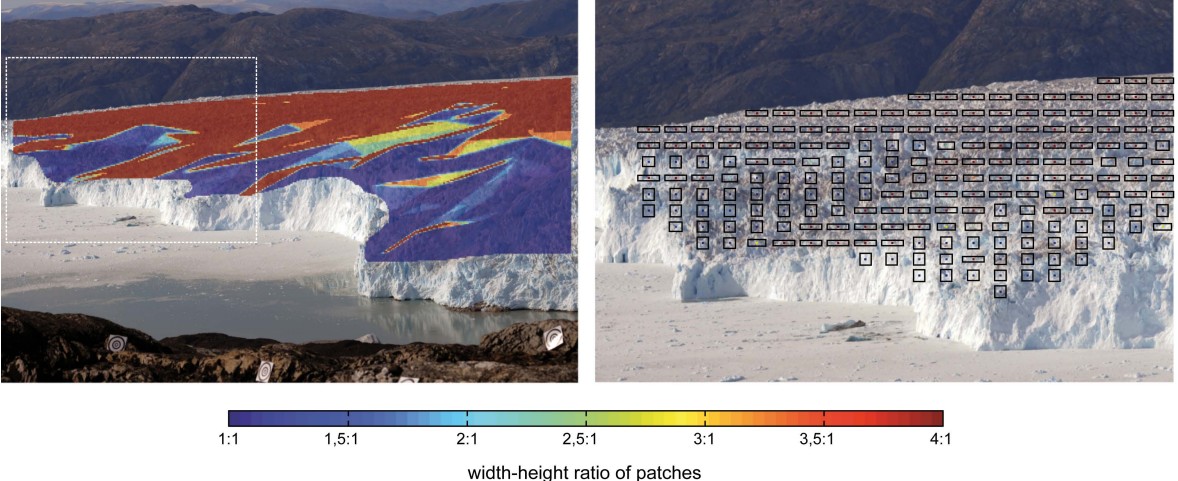

**Figure 3: Locally adaptive patch sizes. The figure shows the result of the automatic patch size definition taking into account distance differences in the patch. On the left, the determined aspect ratios for a patch are shown colour-coded at the corresponding position in the image. For a smaller image excerpt (white rectangle), the resulting patches (black contour) are superimposed on the sequence image on the right.**

For the purpose of glacier motion measurements, the search area can be strongly limited in most cases. It mainly depends on the chosen time interval of the image sequence. Usually, the temporal resolution of the image sequences is high enough to keep the glacier movements between subsequent images small. In most cases the motion direction is predictable and the velocity of a glacier surface point does not change rapidly between subsequent images. Furthermore, the search area also depends on the influence of the camera motion, which can be determined beforehand (see section 2.3.3). This a-priori knowledge can be used to restrict the search area in order to reduce mismatches and ambiguities during tracking.

Depending on the application, different temporal or spatial strategies can be applied for the tracking points in an image sequence. Temporal strategies can be distinguished depending on how images of a sequence are combined for tracking. For instance, a trajectory can be determined by always tracking a point from the first image of a sequence into each of the other sequence images, or it can be determined by successively tracking the point from each image into the subsequent one. The first version in comparison to the second one is stronger regarding error propagation but weaker regarding image de-correlation. Thus, the first strategy might be applied when tracking signalised points - as for camera motion determination (see section 2.3.3) - and the second one when tracking natural points like glacier surface points.

Spatial strategies mean that either a certain feature (e.g. a specific crevasse) can be tracked through the image sequence (Lagrangian approach), or the tracking is performed at a fix position for each image of the sequence (Eulerian approach). The advantage of the first method is that it provides good visual control over the success of the tracking. Advantageous of the second method is that it is independent on the loss of features (e.g. because of calving). This is especially imported for



long-time observations. However, for most applications motion values are required that refer to a fix position in space, which makes the Eulerian approach the standard for the tracking of glacier surfaces. For a subset of the measurement image, Figure 4 shows trajectories resulting from feature tracking on fix position. The single measured translations of a trajectory, which are depicted here arranged one after the other, all refer to the starting point position of the trajectory.

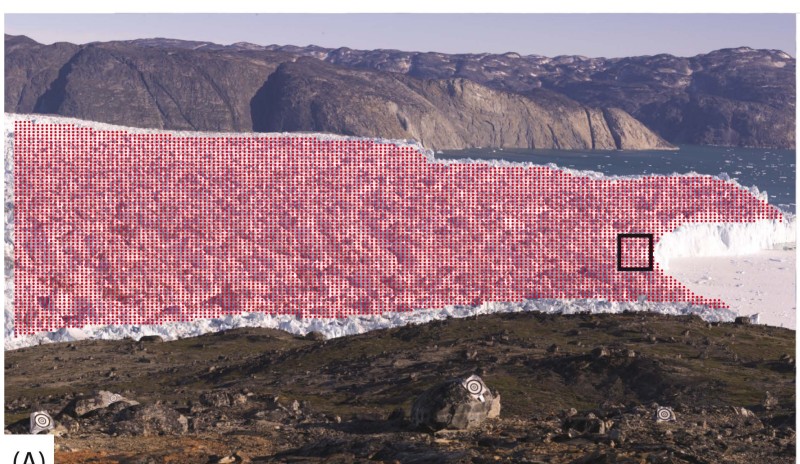
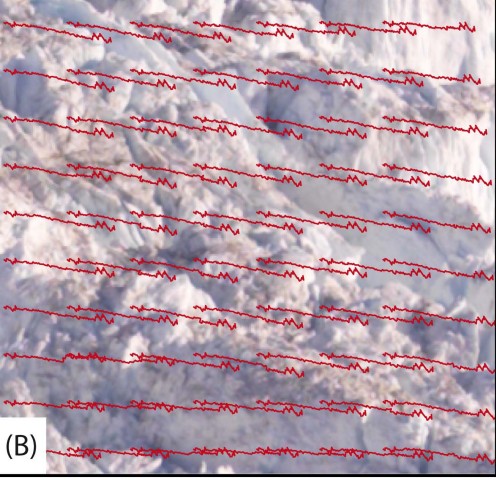

**Figure 4: Raster of measurement points on the glacier surface (A) and resulting trajectories obtained from feature tracking (B). The figure shows the grid of the measuring points (red dots) as a superposition of a sequence image (A). For the black-marked area, the corresponding image section is shown on the right. It is superimposed by the measured trajectories (red lines) for a 24 h**
10 **image sequence (acquisition interval 20 min) (B).**

### 2.3.2 Handling of shadow motion effects

The changing positions of the sun during a day as well as moving clouds generate moving shadows on object surfaces. In case of environmental monitoring time-lapse observation, the motion of shadows may be a rather imported issue. It
15 influences the tracking method described above in such a way that the obtained motion vectors will be a combination of actual glacier motion and shadow motion. More specifically, it is not the whole area covered by a shadow that causes matching errors, but the image areas that change from shadow to non-shadow and vice versa (see Figure 5). Thus, to obtain pure glacier motion the pixels of these areas need to be detected and excluded from the matching (see Figure 5, B).



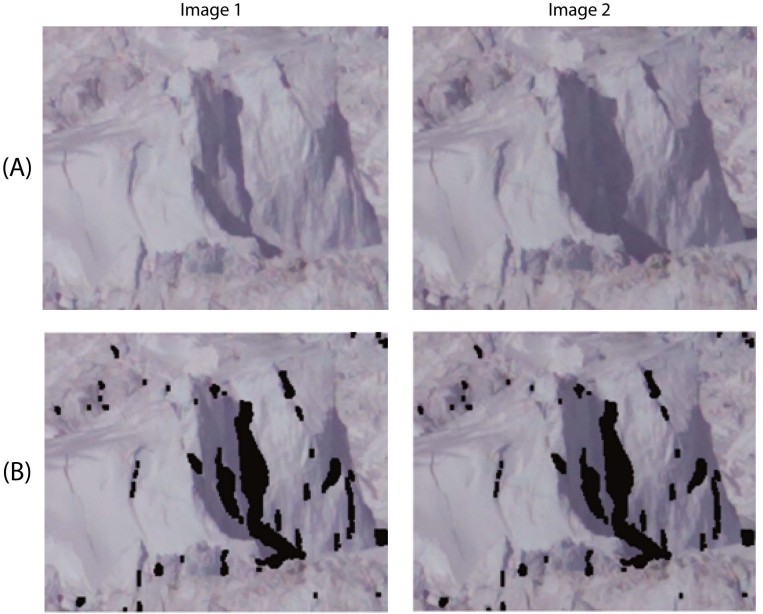

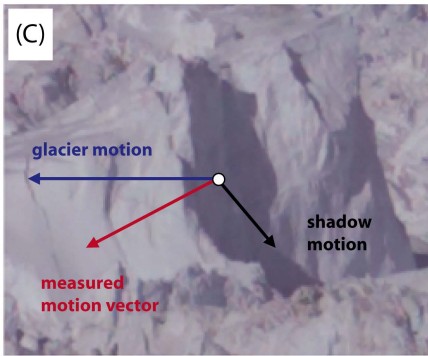

**Figure 5: Influence of moving shadows in image sequences. The figure shows corresponding image sections from two sequence images recorded at a time interval of 30 min (A). In the same image pair those pixels were marked black, which are influenced by shadow motion (B). These pixels falsify the result of the feature tracking because the tracked glacier movement is overlaid by the shadow movement (C).**

The method developed to exclude shadow influenced pixels during the matching will be described with the aid of a synthetic image pair as shown in Figure 6, wherein the slave image is a copy of the master image, but shifted in a predefined way (glacier motion simulation). Additionally, both images are overlaid by a transparent black square whose position in the slave image is shifted by a certain amount in comparison to its position in the master image (shadow motion simulation). In this way a shadow overlaid glacier motion is simulated where reference values for both motion components are available.



**Synthetic image pair:**

master image          search image

simulated glacier motion:

$\Delta x_{G\_Ref}$ = -5,00 pixel
$\Delta y_{G\_Ref}$ = -2,00 pixel

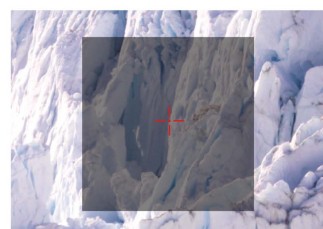
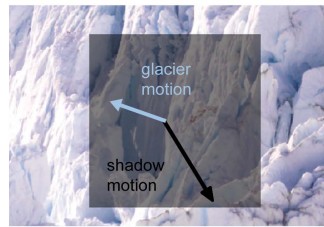

simulated shadow motion:

$\Delta x_S$ = 6,00 pixel
$\Delta y_S$ = -10,00 pixel

**Measurement - LSM with shadow pixel exclusion:**

difference image          excluded pixel

**1. Iteration
(LSM without exclusion):**

measured glacier motion:

$\Delta x_{G\_LSM}$ = -0,98 pixel
$\Delta y_{G\_LSM}$ = -9,52 pixel

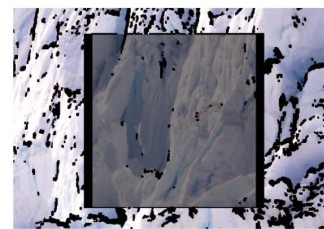

**2. Iteration:
(LSM with shadow pixel
exclusion)**

measured glacier motion:

$\Delta x_{G\_LSM}$ = -4,06 pixel
$\Delta y_{G\_LSM}$ = -5,02 pixel

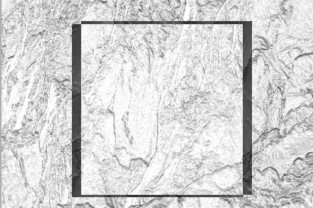
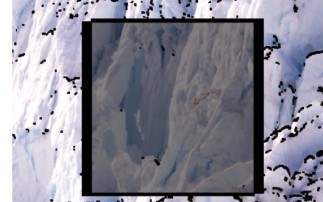

**3. Iteration:
(LSM with shadow pixel
exclusion)**

measured glacier motion:

$\Delta x_{G\_LSM}$ = -5,00 pixel
$\Delta y_{G\_LSM}$ = -2,00 pixel

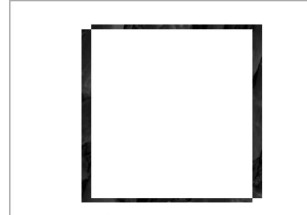
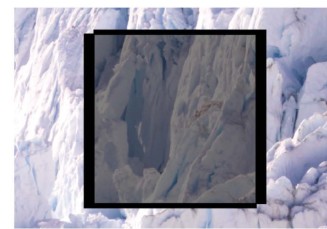

**Figure 6: Least Squares matching with shadow pixel exclusion. The figure shows a synthetically generated image pair, which simulates a glacier motion superimposed by shadow movement, for which reference values are known (top). A method which iteratively detects and excludes shadow motion pixels from the matching is applied to this image pair (below). For each iteration step, the calculated translation parameters (left), the difference image between patch and transformed search patch (center), as well as the resulting exclusion pixels for the next iteration, which are superimposed as black pixels on the patch (right), are shown.**




In order to determine shadow motion pixel, the following iterative approach is applied: Initially all pixels of a patch will be incorporated into the matching process via LSM. The corresponding patch at the thus determined position in the slave image is still completely influenced by the shadow motion. In the ideal case the gray value differences between original and corresponding patch would be zero after LSM transformation. In the first iteration step this perfect match cannot be achieved

because the influence of moving shadows lead to an erroneous position of the corresponding patch and thus to high grey value differences especially at the margins of the shadow areas (Figure 6, 1st iteration). Hence, pixels with grey value differences higher than the single standard deviation are defined as potential shadow motion pixels and are excluded from the matching procedure during the second iteration step. In this way the recalculated position of the corresponding patch is drawn to its real position. After the first iteration not only real shadow motion pixel but also a noticeable amount of non-

shadow pixel is excluded. This ratio improves during the next iteration step along with the improvement of the matching result. The process continues until either the matching result does not change anymore or no more new shadow motion pixels can be found. The comparison of the reference values of the synthetic example in Figure 6 with the tracked glacier motion values after the last iteration step show, that the influence of the simulated shadow motion could be completely eliminated.

Of course, image sequences of natural scenes are influenced by illumination differences as well as by random noise of the sensor. This will also affect the quality of the matching result. Nevertheless the results can be significantly improved applying the method described above (Figure 7).

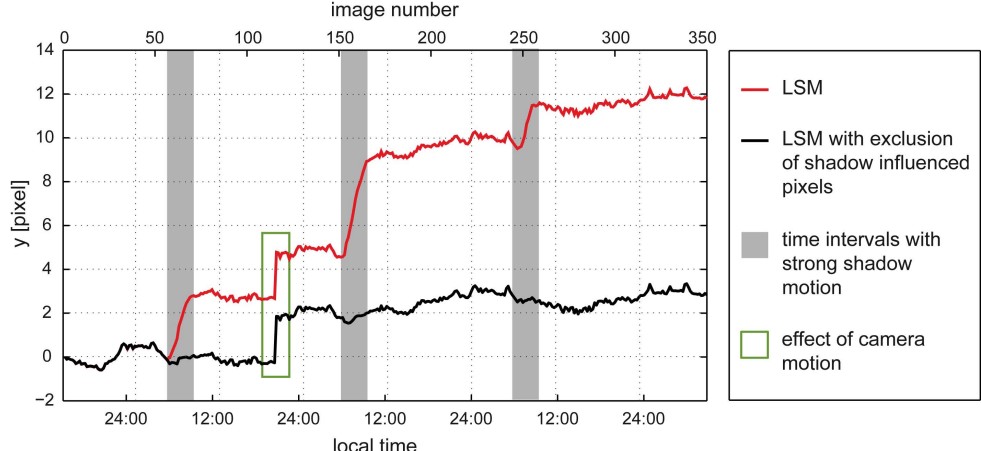

**Figure 7: Drift effects during the measurement of a trajectory due to shadow movement. The figure shows the vertical component of a glacier trajectory measured using least squares matching (LSM) with and without shadow pixel exclusion. At certain times of the day (gray background) particularly strong shadows occur. Both motion curves still contain the influence of the camera motion.**



### 2.3.3 Camera motion

The second main error influence on the motion vectors or trajectories are the effects of the motion of the time-lapse camera itself. Even if a camera system is firmly installed, small movements caused by wind and temperature changes cannot be avoided. Thus, their impact on the measurements needs to be determined and corrected. As techniques such as LSM are

capable of measuring image motion with a precision <0.1 pixel, camera motion has to be corrected for at an equivalent level.

The effects of camera motion can be determined (and corrected) by tracking static points in the images. These can either be signalized targets or natural points (Figure 8). To facilitate camera motion compensation, a sequence image must not only contain the dynamic measurement object (e.g. a moving glacier), but also some terrain in the foreground and/or background.

While signalised points can usually only be installed in the foreground area and are limited in number, natural points may be defined all over static image areas. Although natural points basically provide a better distribution for the modelling of camera motion, their disadvantage is often a lower contrast and their sensitivity to illumination changes, moving shadows and visibility limitations. This will often lead to a lower precision and reliability potential for tracking natural points as compared to signalized targets. However, using the high redundancy of natural points in combination with a robust

estimation method such as RANSAC (Random Sample Consensus, Fischler & Bolles, 1981), this can be compensated to some extent. Thus, for applications where tracking intervals of 24 h (i.e. minimized influence of illumination and shadow variations) are sufficient, natural points might be preferred. For tracking tasks with higher temporal resolution signalised points are often recommendable.

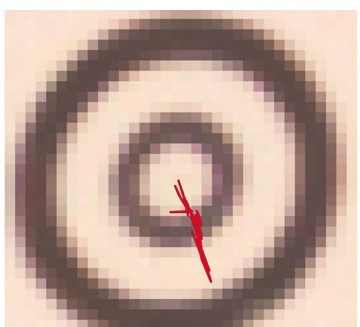
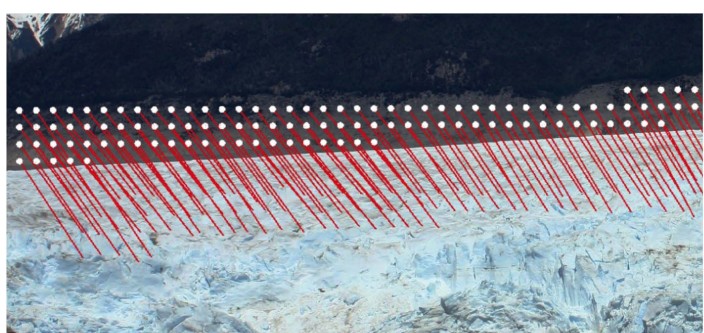

**Figure 8: Tracking of static fiducial marks and/or static natural points. On the left, the figure shows an example of a signalised static point as it is visible in a sequence image. The red line here shows the motion path of the mark during a 24h image sequence (acquisition interval 20 min). On the right, an example for the use of natural static points is shown. The white dots represent a**

**raster of static points that have been defined on the mountain slope in the background of the glacier. The red vectors (enlarged) represent the results for tracking these points from one sequence image into another one recorded 24h later. The red motion lines of the static points (signalised or natural) correspond to the effect of camera motion.**




The tracking of static points results in displacement vectors for each subsequent image pair representing the camera motion between the two consecutive image recording times. These vectors can be used to derive parameters that appropriately describe the influence of camera motion for each individual pixel of the image. In general, two options to model camera motion are possible: The first option is to mathematically describe the real physical motion of the camera in object space

5  defining position and orientation changes of the camera as model parameters. The second option is to mathematically describe the effects of the camera motion in image space by a planar transformation. Applying the first method provides a higher accuracy potential, as also distance depended effects can be modelled, which is not possible when applying a planar image transformation. However, in most cases it is sufficient to describe the effects of camera motion in image space via the two image shift parameters and the rotation parameter of a 2D affine transformation.

In order to calculate the transformation parameters, the start and end coordinates of each tracked static point displacement vector are introduced to the equations of the affine transformation. Since each vector provides two equations, two static points would be sufficient to solve the equation system. Using natural static points, the rate of outliers among the tracked displacement vectors may be significantly higher than for signalised points. However, this can be compensated by using a

15  larger number of the natural points, eliminating outliers by robust techniques such as RANSAC. Applying these transformation parameters to the image coordinates of tracked glacier surface features, an individual correction value can be determined and applied to each of the measured motion vectors (Figure 9).

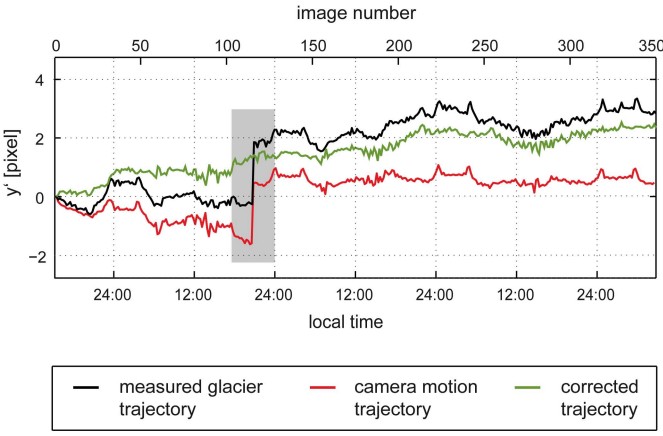
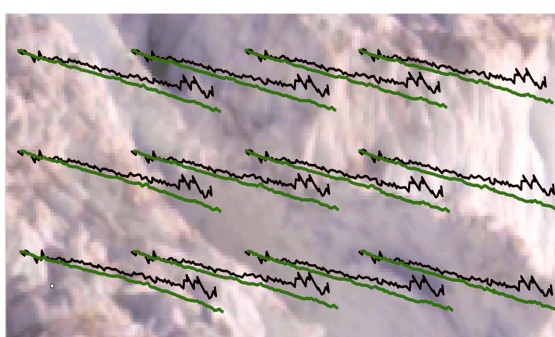

**Figure 9: Application of camera movement compensation. The chart on the left shows the vertical component of a measured trajectory (black), the camera motion trajectory (red) and the corrected trajectory (green). The gray area indicates where a jump in the camera movement occurred. On the right a small subset of a sequence image is shown overlaid by some measured trajectories (tracked trough a 24h image sequence).The originally measured trajectories (black) clearly show fluctuations caused**
25  **by camera motion. In contrast, the corrected trajectories (green) finally represent the smooth glacier movement.**





## 2.4 Geo-referencing

After feature tracking, camera motion and shadow correction, glacier surface motion vectors and trajectories are available in image space. However, to allow for a reasonable interpretation of these motion vectors and trajectories, they need to be scaled from pixel to metric space (Figure 10, A) and linked to their absolute 3D position in object space (Figure 10, B).

5  Obviously, each vector or trajectory has its own scale factor due to their varying distance to the camera. While proper scaling is required for motion analysis, absolute geo-referencing is a precondition to fuse measurements with other data (e.g. remote sensing image overlay) or to compare measurements of different epochs or measurement techniques. This section describes how the geometric reference between image and object space can be realised.

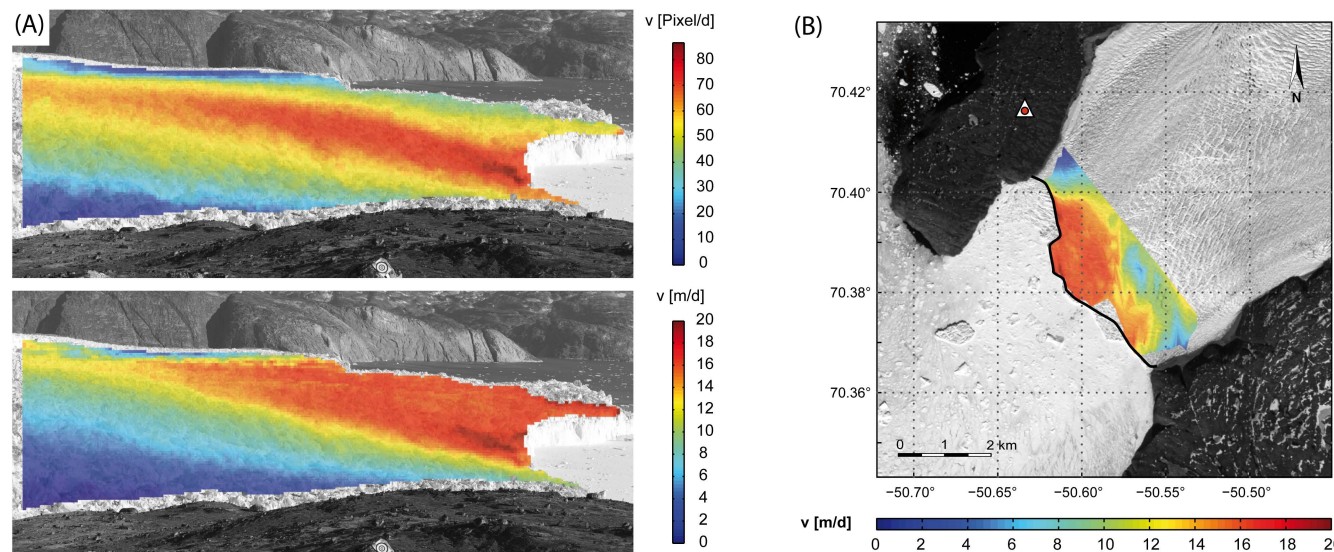

**Figure 10: Result of scaling (A) and geo-referencing (B) The figure shows the color-coded superposition of velocity values derived from measured trajectories. They are shown as unscaled values in image space (A, top) and scaled values transferred into object space (A, bottom), as well as geo-referenced values that can be overlaid on satellite images (B).**

### 2.4.1 Camera orientation and 3D elevation model of the measurement object

The basis for all further evaluation steps is the determination of object coordinates on the glacier surface. These are used to calculate a coarse digital surface model for the glacier area visible in the image sequence, which forms the basis for the determination of the distance between camera and glacier surface points as needed for scaling. For this purpose, further

20  knowledge on the position, orientation and intrinsic calibration of the time-lapse camera is required. In order to determine these parameters we make use of a photogrammetric network setup as described in section 2.2.





The recorded multi-view images, including an image from the time-lapse camera, are processed via photogrammetric bundle block adjustment (e. g. Kraus 2007), allowing for simultaneous determination of camera orientation parameters and glacier surface point coordinates at a high accuracy. The thus obtained glacier surface elevation model is optimally adapted to the time-lapse camera field of view and its oblique viewing angle.

For the processing we predominantly used structure from motion (SFM) tools (such as Agisoft PhotoScan) in combination with our own photogrammetric bundle adjustment library (Schneider, 2008). We took advantage of the SIFT-algorithm in PhotoScan to automatically measure image coordinates of corresponding points in the images of the image block, which were exported into the bundle block adjustment. Going a step further and calculating the sparse cloud while providing the measured camera positions to PhotoScan, the thus determined 3D coordinates of object points can also be exported and used as approximate values for the bundle block adjustment. The major advantage of an open photogrammetric bundle is in the fact that it allows for a thorough error analysis and flexibility regarding different measurement configurations. It can be adapted to different types of control points as well as different sets of camera calibration parameters, scale conditions can be defined, and it provides the possibility to define each variable as fix or parameter to be estimated. Since many SFM tools are rather optimized for fast processing and 3D-visualization than for accurate measurement purposes, some limitations may have to be taken into account, when applying them for measurement tasks. However, when not using a reduced measurement setup it is also possible to determine object coordinates and camera orientation parameters solely using PhotoScan. Preconditions for this are, a network configuration of at least 3 images and 3-5 3D control points.

Due to restrictions of the terrain, the intersection geometry of the photogrammetric network established in the field is usually rather poor. Thus, it is recommendable to measure the cameras positions in the field and to pre-calibrate the cameras in the lab. The cameras camera position coordinates and the parameters of interior orientation and lens distortion are then introduced as fix values into the bundle block adjustment to stabilise the network and to minimize correlations between parameters. Seven degrees of freedom need to be fixed for the network (3 translations, 3 rotations, 1 scale). This can be achieved with at least two 3D control points and one height control point. Thus, the minimum setup, which can be used, consists of an image network of two images with known positions and a single height control point. However to ensure a successful and accurate processing of the photogrammetric network 5-10 control points that are well distributed within the field of view are recommendable.

As result of the photogrammetric bundle block adjustment, the orientation parameters of the time-lapse camera and 3D coordinates of a large number of object points on the glacier surface plus an individual error value for each parameter and coordinate are obtained. In case of solely using SFM tools, camera orientation parameters and very dense 3D point clouds of the glacier surface are derived as well but usually without an integrated error analysis.





### 2.4.2 Distance map

In order to generate the DSM, the calculated 3D object points are meshed to a TIN (triangular irregular network, e.g. Delaunay, 1934). Using the DSM the individual distance between the camera and the corresponding object point on the glacier surface is assigned to each pixel in the time-lapse image. For this purpose, the image ray for each pixel can be

reconstructed based on knowledge about the exterior and interior orientation of the time-lapse camera and intersected with the DSM. From this intersection point and the camera projection centre, the distance value for the corresponding pixel is obtained (see Figure 11). The distance image thus created forms the basis for scaling the individual trajectories and can also be used for automated parameterization of individual patch shapes in image sequence analysis (compare section 2.3.1).

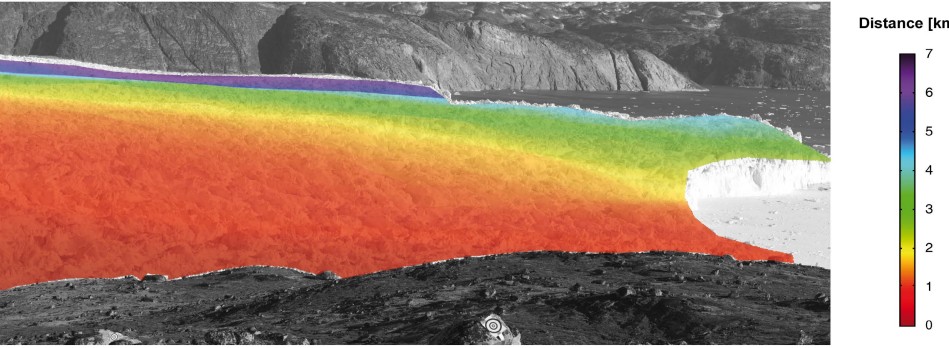

**Figure 11: Distance map**

In case of using PhotoScan, the exterior camera orientation parameters and a depth map can be exported for the time-lapse image. After converting the depth map (containing depth coordinates of object points referring to the camera coordinate

system) into a distance map (containing distance values between object points and the cameras perspective centre), the output can be further used as described in the following.

### 2.4.3 Scaling and position determination

In case of a horizontally oriented camera and an orthogonal viewing direction of the camera to the flow direction of the

glacier, the motion vectors can simply be scaled by the distance values. However, this can usually not be perfectly realised in the field and is often not desirable either, since the sequence cameras field of view should cover the measurement area optimally, which often necessitates an oblique viewing angle. When applying a simple scaling by distance, erroneous vectors (red vectors compared to the correctly scaled vectors in green) would be obtained as shown in Figure 12 (A). The size of the error depends on the motion vectors position in the image and the deviation of the viewing angle from orthogonality to the

glacier motion direction Figure 12 (B).





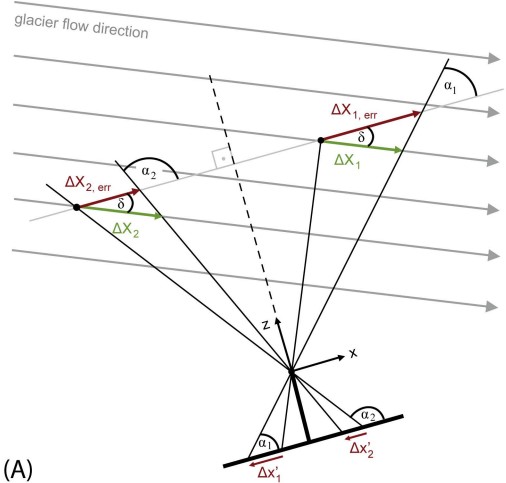

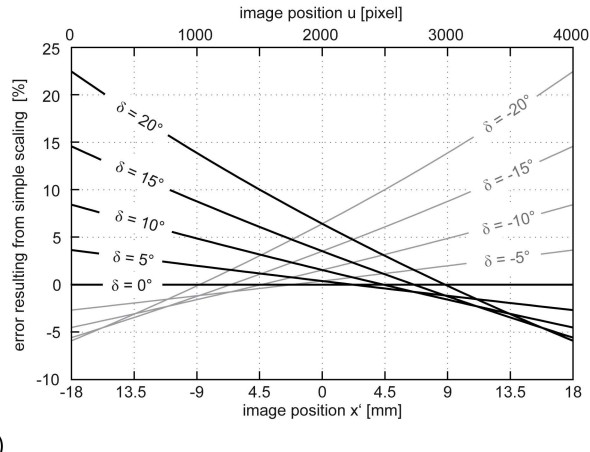

(A)  (B)

**Figure 12: Scaling error in case of a deviation δ from orthogonality between the glacier flow direction and the viewing direction of the camera (shown for the horizontal motion component).**

Thus, a more comprehensive method for the transformation of a vector or trajectory into object space is required. In the monoscopic approach, model assumptions have to be made about the direction of movement of object points. It is assumed that each glacier point moves in a vertical plane oriented along the flow direction of the glacier, i.e. that there is no significant motion across the moving direction (see Figure 13). Starting from the first point P' of a motion vector or

10    trajectory given in image coordinate system, the image ray is reconstructed and the corresponding 3D object point P is determined by means of the known distance. The object point P, the vector of the glacier flow direction and its perpendicular define the vertical movement plane for the measured point. The glacier flow direction can e.g. be obtained via flow-line patterns that are visible in satellite orthophotos. By intersecting the image ray of the second motion vector point or of each further trajectory point with the thus defined vertical plane, the 3D object coordinates can be determined for all points of a

15    motion vector or trajectory.

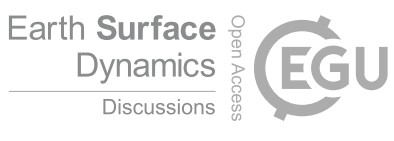



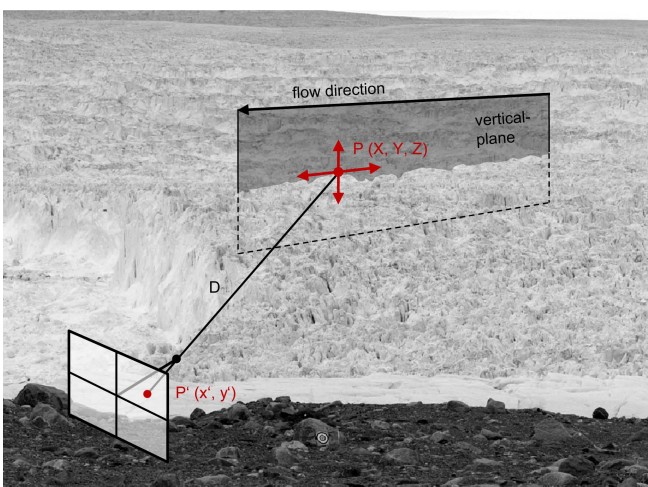

**Figure 13: Transformation into object space.**

## 2.5 Aspects of accuracy

The error of a determined motion vector is composed of several individual error influences. According to the individual steps of the image sequence analysis, four main error sources can be distinguished: The error of the image point tracking, the error of the camera movement correction and the scaling error as well as the error of geo-referencing. Whereas the three first mentioned errors have an influence on the accuracy of the determined translations, the geo-referencing error affects the accuracy of the spatial reference of the translation values. The errors of image point tracking and camera movement

correction occur during the measurement in image space, the errors of scaling and the geo-referencing while transforming the measurements into object space. The effect of each of these errors on the measured translations is also influenced by parameters of the time-lapse measurement setup, namely the principal distance and sensor size of the camera, the recording interval of the time-lapse sequence, the distance to the object, the local position of the measurement point in the image and the amount of the measured translation itself.

For a sample trajectory from an image sequence measurement at Jakobshavn Isbræ in May 2010, the main error effects mentioned above have been estimated for each translation value of the trajectory applying statistical error estimation techniques (compare i.e. Niemeier, 2002; Taylor, 1997):

For the determination of the error of image point tracking the error analysis that is integrated part of the LSM procedure provides accuracy values that result from the statistical evaluation of gray-value differences between two patches. The standard deviation for a matching is thus a measure of the influence of the sensor noise and how accurately the transformation model of the LSM approximates reality. Although the influence of shadow motion is significantly reduced by





the method explained in section 2.3.2 a remaining error of shadow motion influence and in case of the vertical translation component also refraction effects have to be considered additionally. Thus, on average, an error of 0.05 pixel for the horizontal and 0.17 pixel for the vertical motion component were estimated for the single translations of the trajectory. These errors in image space translate into error values of 2.3 and 8.1 cm in object space referring to the distance of 3000 m between measuring point and time-lapse camera.

The accuracy with which the camera motion correction can be applied is influenced by the tracking error of the static targets as well as by the quality of the functional model, which is used to mathematically describe the camera motion (see section 2.3.3). On average, we estimated an error of camera motion correction of 0.14 pixels for the measured translations of the example trajectory. Regarding to the distance of the measuring point from the camera, this corresponds to an error of 6.8 cm for each motion component.

The error of scaling is influenced by the scale error and inner accuracy of the photogrammetric network as well as by the error of the angle that describes the deviation from the orthogonality of camera direction and glacier movement direction. The latter is strongly dependent on the position of the measuring point in the image (section 2.4.3) and mostly influences the horizontal motion component. The scale error of the photogrammetric network depends on the accuracy of the measured camera or baseline positions and thus on the chosen geo-referencing measurement setup (see section 2.2). The inner accuracy of the photogrammetric network depends on the network configuration (convergent angles between image views, number and distribution of image positions), the quality of camera calibration and the cameras sensor resolution. The total scaling error was estimated with 7.27% for the horizontal and 0.34% for the vertical motion component.  For a measured motion component of 0.5 m this corresponds to a scaling error of 3.6 cm (horizontal) and 0.2 cm (vertical).

These individual errors were propagated into a mean total error of 9.2 cm for the horizontal translations of the example trajectory and 10.7 cm for the vertical translations. Note that the major error components are of absolute nature; that means that the relative error (i.e. error divided by velocity) will almost linearly drop with increasing velocity. It should also be noted that the camera movement and geo-referencing related error components show a local and temporal correlation; that means that spatial or temporal velocity gradients will have a higher precision, as correlated error effects will cancel out.

These error calculations exemplarily described above for a single trajectory were conducted for each trajectory of the measured area on the Jakobshavn glacier. Figure 14 (A) shows the horizontal daily velocity values derived from the measured trajectories and the corresponding error values for the velocities calculated from the trajectories translation errors. Figure 14 (B) shows velocity errors of 5 to 15 cmd$^{-1}$ and the decrease of the accuracy with increasing distance between measurement point and time-lapse camera.



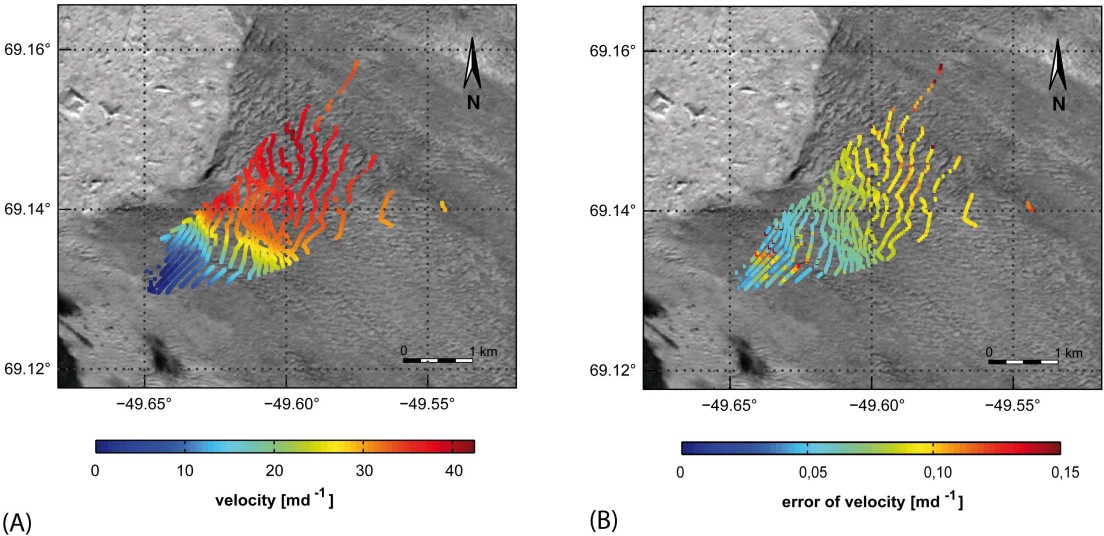

(A)                    (B)

**Figure 14. Horizontal velocity field (A) and corresponding error values (B) (example: Jakobshavn Isbræ, May 2010).**

A further issue to consider is the error of the geo-referencing which does not influence the measurement values themselves but is important when the measurements need to be compared with other data sets. This error strongly depends on the chosen method and instrumental setup for the geo-referencing (compare section 2.2). Generally we need to distinguish between lateral errors and height errors. Thereby the lateral accuracy of the spatial reference points of the individual trajectories and

motion vectors is influenced by the position errors of the baseline (formed by the positions of the cameras of the photogrammetric network or by the positions of the total station measurements). The error of a height control point causes a rotation error around the baseline and thus a height error of each measuring point. The error of a far-distant height control point thereby has a minor effect on the measurement points than the same error of a height control point with a smaller distance to the baseline.

**3 Assessment of glacier motion fields: four case studies**

The velocity fields derived by monoscopic image sequence analysis can be further used for a wide range of motion analyses tasks. Depending on the application and characteristics of the measurement object, the basic method as described above may be individually adapted. Since we are focusing on glaciology here, this section will present examples for different glaciology applications of the method to show its potential. In comparison to satellite based analysis, terrestrial photogrammetric

techniques cover only small areas, but provide measurements at a very high spatio-temporal resolution and accuracy. Thus, the method is especially suitable for the investigation of objects, which show a high motion dynamic. The derived motion



vectors or trajectories contain information about horizontal as well as vertical motion components, allowing for manifold glaciologic analyses.

## 3.1 Horizontal glacier motion

Horizontal glacier motion fields derived from time-lapse imagery provide information about the motion of the glacier as well as spatial and temporal variations of velocity. They have proven to be a valuable database especially for the investigation of fast-flowing, high-dynamic glaciers, like many outlet glaciers of the big ice sheets.

### 3.1.1 Glacier motion velocity fields

Knowing the recording interval of the time-lapse images, velocity fields can directly be derived from the glacier motion fields. Depending on the time interval of the time-lapse series, velocity fields can be generated with very high temporal resolution. They are especially suitable for very fast moving glaciers such as many of the Greenland outlet glaciers that are of high scientific interest in the context of mass balance determination and sea level rise prediction. For these glaciers - with maximal velocities in the order of several tens of meters per day - the method is a useful supplement to glacier flow velocities derived by remote sensing methods which are hampered by their revisit time intervals. Figure 15 (A) shows velocity fields close to the front of Jakobshavn Isbræ (West Greenland) obtained by monoscopic terrestrial image sequence analysis over six years, displaying glacier surface velocities in the order of 40 meters per day. Each velocity field consists of up to 4000 single measurement values. Similar velocities have also been determined by 3D feature tracking in multi-temporal terrestrial laser scanner data (Schwalbe et.al. 2008).

Another advantage of the high temporal resolution of the terrestrial image sequences is that this allows for filtering the data. If e.g. daily velocities are required but the time-lapse camera is operated with a 10 to 20 minute time interval, the redundancy in the measurement values can be used to detect and eliminate outliers and to improve the precision by averaging. Thus, also investigations of glaciers with lower velocities may benefit from the terrestrial time-lapse measurements. This may especially be important for regions with changing weather conditions and frequent rainfall or fog as it is for instance the case for Patagonian glaciers. As an example Figure 15 (B) shows the velocity field of Glacier Grey (Southern Patagonian Ice Field) where each velocity value has been derived from about 300 single motion vector measurements (Schwalbe et.al. 2016).




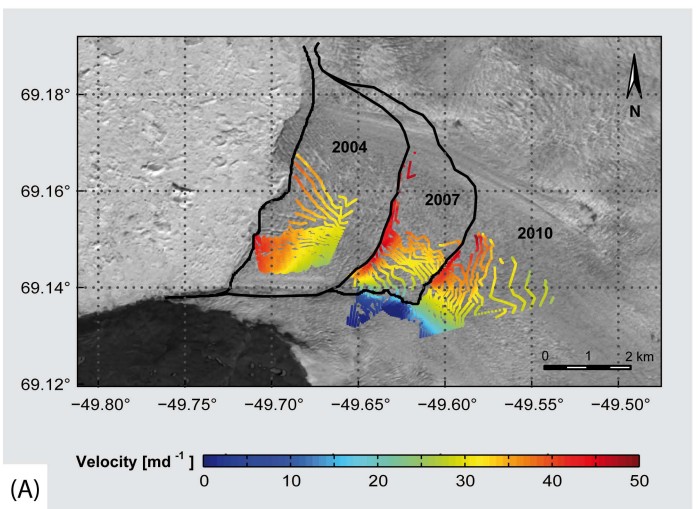

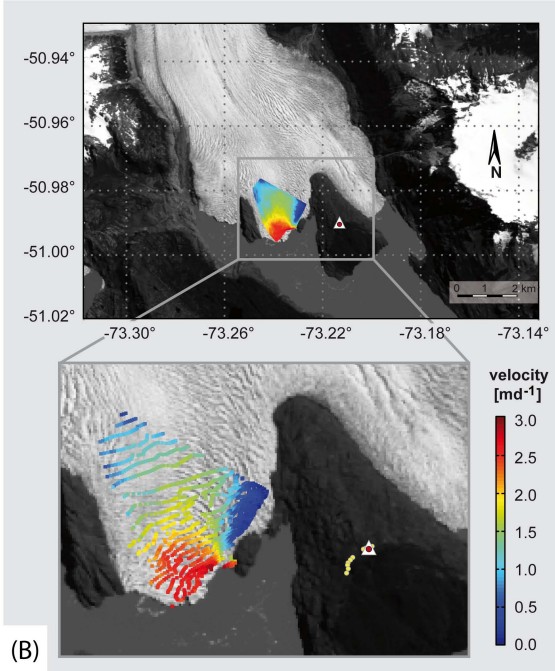

**Figure 15: Velocity fields of Jakobshavn Isbræ, West Greenland (A) and Glaciar Grey, (Southern Patagonian Ice Field (B))**

### 3.1.2 Glacier motion during calving events

Due to the (almost arbitrarily) large repetition rates, the analysis of terrestrial image sequences also allows for the investigation of short-term changes in the velocity field during calving events. Calving events take place within a short period of time, thus requiring a high temporal resolution of the measurement method. On the other hand, the beginning of a calving event can hardly be predicted, which makes it necessary to autonomously record the glacier front area over a long period of time. As an example for this kind of application Figure 16 shows the velocity field during a large calving event at

Jakobshavn Isbræ (West Greenland) on the basis of an image sequence over several weeks in May / June 2010 recorded with a temporal resolution of 20 minutes.

During the calving event, the motion velocity close to the glacier front locally jumped towards up to 70 md$^{-1}$ within a period of a few hours. The velocity thus almost doubled in the area of the new glacier front. The velocity increase decreased with

increasing distance to the glacier front, but the impact of the calving event on the motion behaviour of the glacier could still be determined up to 1 km upstream. The rapid velocity increase during the calving was followed by a slower decay of the motion rates over a period of 4-5 days. Figure 16 shows the change of the velocity before and after the calving event for three positions at different distances to the glacier front. Further analysis methods and results with regard to the investigation of calving events are described in detail in (Rosenau et. al., 2013).





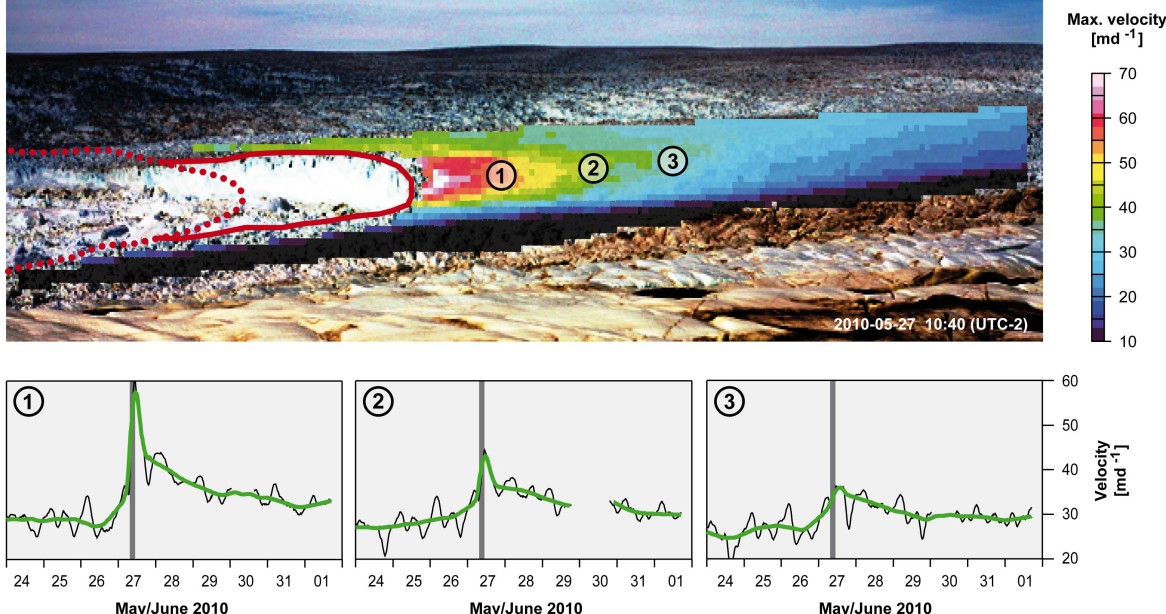

**Figure 16: Glacier flow velocities during a calving event at Jakobshavn Isbræ (Rosenau et. al., 2013). The figure shows the maximum velocities occurring during a calving event as a colour-coded overlay of the measurement image (top). The position of the glacier front after the calving event is depicted as a red line, its position before calving as a dotted red line. For the positions 1-3, the velocity changes during the calving event are shown below. The vertical gray line marks the time of the main calving. The black curve represents the unfiltered measured values; the green curve represents the filtered measurement values.**

## 1.1 Vertical glacier motion

The vertical component of the glacier motion field can be especially useful to draw conclusions on events taking place underneath the glacier surface and to recognize specific features that lie within or under the glacier. The vertical movement field thus has the potential to give an indirect view into the glacier.

### 1.1.1 Grounding line determination

When investigating outlet glaciers that terminate into the sea, the vertical motion component of the trajectories may provide information about tidal induced movements of the frontal area of the glacier tongue. Figure 17 (A) shows the vertical motion component of a single trajectory compared to tidal height measured by a pressure gauge. Using tidal models or sea level measurements as a reference, various parameters such as vertical tidal participation, phase shift and inclination can be estimated for the measured trajectories. These parameters characterize the differences between the movements of the glacier surface compared to the tidal curve. In particular, the scale factor in vertical direction, which can be interpreted as a damping factor of the amplitude, can be used as an indicator for the detection of floating glacier areas (Figure 17, B). The detailed





procedure for calculating the percentage of the vertical glacier movement compared with the tidal range (free float) is explained in (Dietrich et al., 2007). The transition from the free float section of the glacier (with almost 100% tidal movement participation) to the grounded section (with no tidal movement participation) can be used for the determination of the glaciers grounding line. Multiple time lapse measurements at Jakobshavn Isbræ in 2004, 2007 and 2010 even allowed for

5    the documentation of the migration of the glaciers grounding line (Figure 17, C) (Rosenau et. al., 2013).

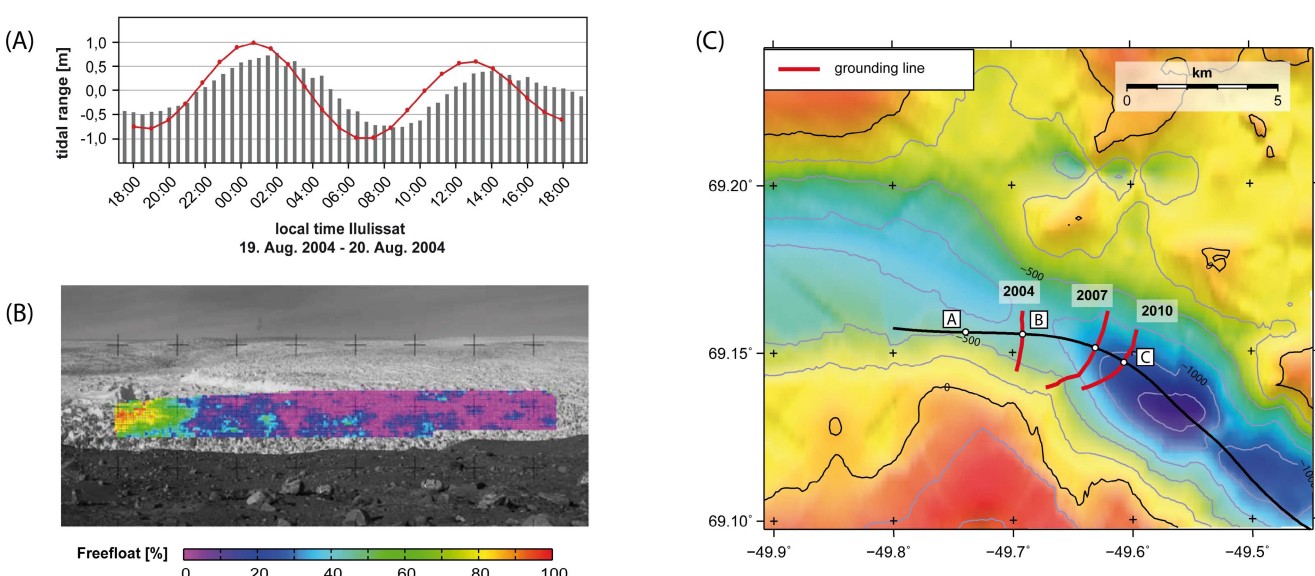

**Figure 17: Free float measurements (left, from (Dietrich et al., 2007)) and grounding line determination at Jakobshavn Isbræ (right, from (Rosenau et.al., 2013), background subglacial topografie from (Plummer et.al., 2008)).**

### 1.1.2 Recognition of sub-glacial drainage channels

Another example for the possibility to obtain information about sub-glacial processes from vertical motion fields derived from terrestrial image sequences is the detection of a sub-glacial channel during a glacier lake outburst flood (GLOF) event. During a GLOF event, a glacier margin lake spontaneously starts to drain underneath the glacier, causing flash floods in

15    downstream valleys. The event is triggered by the opening of a sub-glacial channel that drains the lake water and is quickly widened by erosion and melting. A typical GLOF event may take a few hours to days. After the outburst, the channel soon collapses and the lake refills until a new channel opens up and the cycle starts again (see Clague & Evans, 1994; Tweed & Russell, 1999; Dussaillant et al., 2010).

20    A time-lapse camera observing the water level of glacier margin lakes may be used as a GLOF early warning system if sudden water level changes can be detected reliably in the image data by a suitable image processing procedure (Mulsow et al., 2014). Vertical motion patterns of the glacier surface obtained from the same image data may depict a suitable basis to




localize such a sub-glacial channel and to investigate the development of the channel during and after a GLOF event. As an example, Figure 18 shows vertical glacier motion fields of the Colonia glacier (Northern Patagonian ice field), that deliver information about the glaciers behavior during and after a GLOF event of its glacier margin lake Cachet II. The high temporal resolution of the motion fields allow for an analysis of the tunnel formation and collapse in relation to the changing

5    water level of the lake obtained from a GLOF early warning camera (Figure 18, A). In order to localize and visualize the drainage channel, daily subsidence fields of the glacier were integrated over a period of several days and projected into a satellite image (Figure 18, right). A more detailed explanation of the accordant measurements and the applied method is given in (Schwalbe. et. al. 2016).

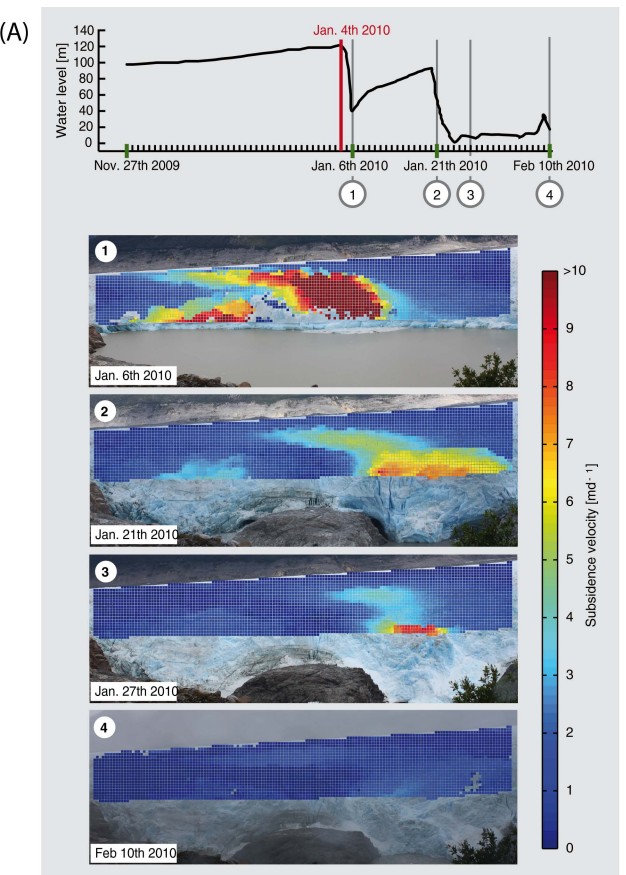
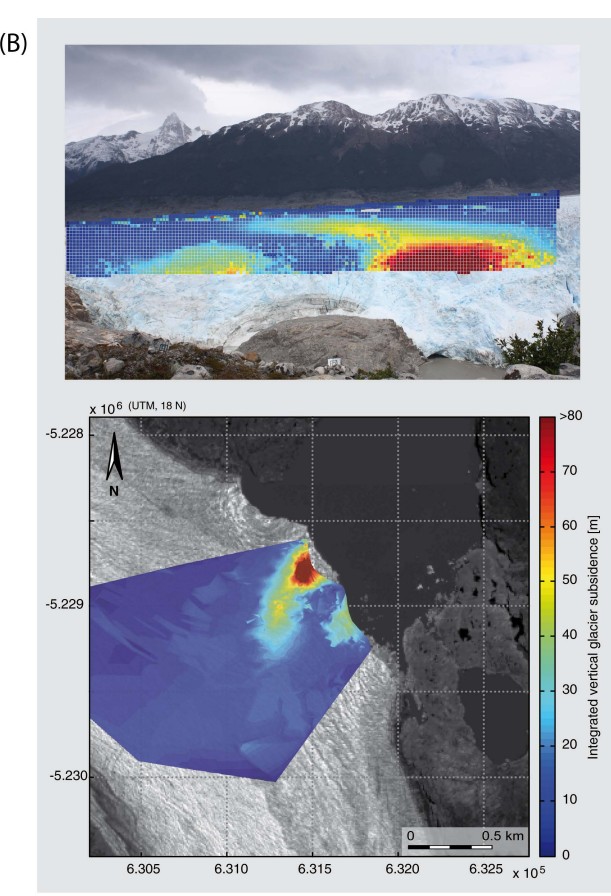

**Figure 18: Documentation of the formation (A) and localization (B) of a sub-glacial drainage channel at Colonia glacier (Northern Patagonian ice field) during a glacier lake outburst event of Lake Cachet II.**



## 2 Conclusion

The presented approach allows for the analysis of monoscopic image sequences in order to derive glacier motion vector fields with high spatial and temporal resolution based on tracking a large number of glacier surface points. These motion vector fields are of great value for the analysis of glacier motion behaviour particularly of glaciers showing high motion

dynamics. Terrestrial photogrammetric time-lapse image measurements depict a powerful tool for providing glacier motion data with high accuracy at a high spatial and temporal resolution. Compared to other measuring methods, which are used for the observation of glaciers, automatic image sequence analysis also offers the advantages of a high flexibility and versatility at rather low instrumental cost. Obviously, there is the disadvantage that terrestrial measurement methods are not suitable for investigations of large areas. They are therefore to be considered as complementary methods to measurement methods based

on aircraft or satellite imagery, which have the potential of covering large areas, but at a limited temporal resolution.

The methodology of monoscopic image sequence analysis consists of the determination of glacier surface point motion vectors and trajectories in image space and the transformation of the motion vectors into object space. The trajectories of the glacier points determined by sub-pixel accuracy image matching techniques may be distorted by migrating shadows on the

glacier surface and by instabilities of the camera. The influence of the shadows can be eliminated during image point tracking by a technique of iteratively excluding shadow-influenced pixels from image matching. The influence of camera movement can be compensated by tracking fixed targets visible in the image sequences. In order to scale the trajectories from image space to metric measurements, the accordant distances between the camera and the according object points are required, which can be provided by a photogrammetric network adjustment approach. For the purpose of geo-referencing,

different measurement setups are possible, which can be chosen depending on whether the focus of the measurement is on high positioning accuracy or light-weight measurement equipment and flexibility in the field.

The error analyses of the trajectories derived from the image sequences showed that the individual motion values can be determined with accuracy in the order of several centimetres for glacier surface points at a distance of several kilometres

from the camera. The method allows for the measurement of both horizontal and vertical motion components. The described practical examples have shown that terrestrial time-lapse measurements can provide a suitable data basis for a wide spectrum of applications in glaciology. The determination of horizontal motion fields for fast flowing glaciers and the investigation of calving events have shown, that terrestrial time-lapse measurements may be a valuable supplement to satellite based observations of glaciers. While satellite images can cover large glacier areas, the terrestrial method takes advantage of its

high temporal and spatial resolution and provides data for special local investigations of glacier areas which are characterized by highly dynamic changes. The studies for the detection of a glacier's grounding line or the localization of sub-glacial channels during GLOF events have shown, that vertical motion data of glacier surfaces obtained from terrestrial time-lapse measurements can be used to indirectly draw conclusions about events occurring within or underneath a glacier.





While the method of image sequence analysis as described here is optimized for the application for the motion analysis of glaciers, the use of the approaches is also conceivable for manifold other environmental investigations, e.g. motion analysis of landslides or even flow measurements of rivers.

5 **Acknowledgment**

The work presented in this paper has been supported by the German Research Foundation (MA 2504/5-1, MA 2504/14-1), by the German Federal Ministry of Education and Research (BMBF International Office) and by the support program "support the best" (StB) of the TU Dresden. Software modules for glacier surface point tracking and geo-referencing will soon be made available on the author's webpage.

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
