# Peer review of "Determination of high resolution spatio-temporal glacier motion fields from time-lapse sequences"

_Earth Surface Dynamics, 2017_

## Referee Comment (RC1) · Anonymous Referee #1 · 5 Jul 2017

The authors present a summary of their methodological developments in terrestrial photogrammetry for glacier monitoring. Unfortunately, most of the material in the manuscript has already been presented in a previous paper by the authors (Rosenau et al. 2013). Since the authors provide the reference to their previous study only on page 23 I was left with the impression to read through original material and spend the better part of a day on this review!

Detailed comments:

The introduction provides a historical perspective on the use of terrestrial photogrammetry for glacier studies but I feel it leaves out some significant recent advances in the

field of environmental monitoring with photogrammetric methods including the monitoring of glacier (e.g. Messerli and Grinsted 2015), landslides (e.g. Travelletti et al. 2013) or discharge monitoring (e.g. Stumpf et al. 2016). Some of those tools are readily available in the public domain and can be used for the proposed applications. It would therefore be helpful to explain briefly possible shortcomings of available methods and how the proposed method addresses those issues.

Addtitional references:

Messerli, A. and Grinsted, A.: Image GeoRectification And Feature Tracking toolbox: ImGRAFT, Geosci. Instrum. Method. Data Syst., 4, 23-34, 2015, doi:10.5194/gi-4-23-2015

Travelletti, Julien, et al. "Correlation of multi-temporal ground-based optical images for landslide monitoring: Application, potential and limitations." ISPRS Journal of Photogrammetry and Remote Sensing 70 (2012): 39-55.

Stumpf, André, et al. "Photogrammetric discharge monitoring of small tropical mountain rivers: A case study at Rivière des Pluies, Réunion Island." Water Resources Research 52.6 (2016): 4550-4570.

p.3, l.12: 'translation into object space' Maybe 'transformation' or 'mapping' would be more adequate than 'translation' here

p.5; l.9: "This can be achieved by establishing a local photogrammetric network (consisting of several convergent images taken from different positions as shown in Figure 2)"

As I understand this, this authors suggest a single 3D reconstruction of the glacier surface for a single time steps as opposed to the time-lapse camera which will acquire time-series for the motion measurements? This implicitly comprises the hyphothesis that the glacier surface topography will not undergo large changes during the monitoring period. In particular for the observation of calving event this appears as a rather

strong assumption which should be explained in greater detail.

General comment on 2.2 Measurement Setup: While the section clearly describes several options for the measuring setup and states requirements (e.g. "should be replaced by differential GNSS measurements." "an elevated camera position is required", "it is necessary to have static points", "it is recommendable to use redundant information in the process of geo-referencing"). While I agree with those considerations it is still not clear for me which (similar or different setups and strategies were used in the study. I would encourage the authors to provide more explicit information on the actually realized setups.

p. 8: "Thus, the first strategy might be applied when tracking signalised points" Similar to my previous comment, please avoid hedging, and state what has been actually done.

p. 12, l.7: 'higher than the single standard deviation' The standard deviation of the gray values within the search patch? This automatically assign several pixels as shadow pixels even if not shadows are present at all. It seems to me that on a glacier this might mask many areas with important texture and could leave you only with the areas that are difficult to match? Please comment.

p.13, l.2: "error influence on the motion vectors" This sounds a bit awkward. I suggest something like 'error source that might impact'...

p.14, l.9: "two image shift parameters and the rotation parameter" If you are only considering those two parameters you will estimate a rigid transformation and won't need to invoke an affine transformation. Please clarify.

p.16, l.10: "Going a step further and calculating the sparse cloud while providing the measured camera positions to PhotoScan, the thus determined 3D coordinates of object points can also be exported and used as approximate values for the bundle block adjustment."

I do not understand the purpose of this step. Errors in the internal and external calibration parameters plus matching errors will propagate into the sparse point cloud. Running another bundle adjustment with the resulting 3D points will hence certainly suffer from those errors? Please clarify.

Similar too my previous comments p.16 comprises a lot of ambiguity. Some example: "we predominantly used structure from motion (SFM) tools we predominantly used structure from motion (SFM) tools (such as Agisoft PhotoScan)"

Which other SFM tools where used and for what?

"It can be adapted to different types of control points as well as different sets of camera calibration parameters, scale conditions can be defined, and it provides the possibility to define each variable as fix or parameter to be estimated."

Which type of cameras and camera calibration models where used? How did you adapt the tool and the free parameters and based on which criteria?

"Since many SFM tools are rather optimized for fast processing and 3D-visualization than for accurate measurement purposes, some limitations may have to be taken into account, when applying them for measurement tasks."

Which tools are you talking about? Can you back this with numbers or previous studies?

"However, when not using a reduced measurement setup it is also possible to determine object coordinates and camera orientation parameters solely using PhotoScan."

What is a reduced measurement setup? Is this an option or something you have done in this study?

"Thus, it is recommendable to measure the cameras positions in the field and to pre-calibrate the cameras..." I agree, but it is again not clear if that is a recommendation or something you implemented in this study.

I there are more examples like this.

p.17, l.13: "In case of using PhotoScan, the exterior camera orientation parameters and a depth map can be exported for the time-lapse image."

Above you explained that you use you own in-house bundle adjustment and in combination with pre-calibration of the internal parameters. Why would you then export these parameters from Photoscan?

p.18, l. 12: "The glacier flow direction can e.g. be obtained via flow-line patterns that are visible in satellite orthophotos.

Does this imply that the glacier flow direction is constant across the scene (as shown in Fig 12A)? This would be a rather strong assumption for observations in areas where the glacier flow is bending with the topography. Please clarify and state possible limitations that may arise from this assumption.

p.19, l. 16: "For a sample trajectory from an image sequence measurement at Jakobshavn Isbræ in May 2010"

I feel it would be helpful to first introduce the study sites with their respective measurement setups before providing results on the accuracy. At this point it is rather difficult for the reader to understand if the error budget estimation is representative for all sites.

p.20, l.20: "These individual errors were propagated into a mean total error of 9.2 cm "

It might be useful to provide the formula for the error propagation. Why did you not also assess the measured velocities against independent in-situ measurements?

General comment related to section 3 Assessment of glacier motion fields: four case studies.

I feel that a lot of important information regarding the case studies is missing including maps of the measurements setup, the type of deployed cameras, the duration and frequency of the acquisitions of monoscopic images as well as stereo views. Introducing the particularities of the study sites might also help the readers to understand

which specific choices were made following the rather generic description in section 2.2. Another interesting aspect that might deserve some further considerations is the operation of the camera systems in the those rather harsh environmental conditions (e.g. power supply, data storage and submission, etc.).

3.1 Horizontal glacier motion

Is the glacier motion in the presented case studies purely horizontal? If not your measurements may also comprise a vertical component. Please clarify.

3.1.1 Glacier motion velocity fields

Considering all the effort for setting up a time-lapse system it is more than unfortunate that you only present the aggregated means. Surely you derived some interesting time-series that you could present? What can we learn from those time-series regarding the process of glacier flow? Section 3.1.2 is much more mature in this regard.

p. 23, l 14: "velocity increase decreased with increasing" There is a lot of increase and decrease here. Please reformulate.

Figure 16: I would be helpful to provide a scale for the upper figure (e.g. distance between to arbitrary points on the stable terrain). Please also provide the details on the smoothing filter (e.g. rolling mean with a window size of x?) used.

1.1 Vertical glacier motion Something went wrong with the section numbering here.

1.1.1 Grounding line determination

There are several issues in this section.

The caption of Figure 17 refers to only two of the 3 subfigures and raises the impression that all the results presented originate from other studies. The letters A-C used to number the subfigures are not used at all. In Fig 17 A it is not clear which measurement is from which source. The y-axis gives the impression that both curves show the tidal range while one of them shows (I assume) the vertical component of the glacier motion.

"Figure 17 (A) shows the vertical motion component of a single trajectory compared to tidal height measured by a pressure gauge."

Neither from the text nor in the figure it is clear which measurement is which.

"In particular, the scale factor in vertical direction" The paper is not really self-contained here. Please explain how the scale factor is computed.

"Multiple time lapse measurements at Jakobshavn Isbræ in 2004, 2007 and 2010 even allowed for the documentation of the migration of the glaciers grounding line (Figure 17, C) (Rosenau et. al., 2013)."

I finally had a look at the study by Rosenau et al. 2013 and came to realize that an important part of the presented material has already been published in this previous paper by the authors. This includes the measurements on the Jakobshavn Isbrae glacier, the corrections for camera movement, the projection into object-space, estimation of the error budget, the delineation of the grounding line, and the documentation of the calving event. I stop the review here.

---

## Referee Comment (RC2) · Anonymous Referee #2 · 7 Jul 2017

I have read the paper with interest, and I think this is a potential interesting contribution for the special issue as it matches very well with the topic. My main concern about the paper is the very large similarities with the paper already published by Roseneau et al. 2013. Indeed this paper is not cited in the introduction when I think that it should be clearly stated that this paper exists and which is the exact contribution and adding of this new paper. It would also allow to prepare a revised version more focussed and shorter, as in its current state it seems to me that it results a little bit sparse and too long (18 figures is probably excessive for a paper).

As a suggestion, I think that the paper will be more interesting for readers focussing in
the detailed presentation of the methodology and validation that you do that exceeds the one offered in Roseneau et al.2013. Much of the sections dealing with applications and results could be removed and synthetized in a discussion section. In this way, I think it would be useful to add a figure showing a scheme with the complete workflow applied to the figures in a sequential way. I would avoid to exceed 10-12 figures in total.

Minor comments:

-In general the methodology is presented in a clear way and it is possible to be understood by the majority of readers, but some sentences could be simplified to facilitate the lecture for people not very familiarized with this technique. An example is. "We limit ourselves to monoscopic image motion capture and processing delivering two-dimensional velocity field information here, as the 15 glaciology phenomena observed in the practical experiments do not show significant across-track motion and can thus be well described in 2D."; or to explain better some concepts as "decorrelation" used in page 2 line 29.

- It is necessary to read more than half of the paper to realize that you are using Photoscan and a library developed for yourselves for the analyses. I would mention this earlier in the paper and I would indicate if the library is open for all users. - P7 line 19: First, not fist. - P6, line 23. " The cameras camera position...." - How is corrected the distortion of the lens? Using the info provided by the producers or using some tool as the one provided by Photoscan? Does t - If I understood well, all the analyses were made with only one camera. In Figure 2 the inclusion of more photogrammetric cameras may lead to confusion. - Please check well the references-citations as there are several mistakes. Indeed the reference of the paper Roseneau et al. 2013 is wrong.

Hoping my comments will be useful.
* * *
**ESurfD**

---

## Author Comment (AC1) · 31 Aug 2017

**Answers to Anonymous Referee #1**

We want to thank the referees for their comments and suggestions and for the time they spend reviewing our manuscript. Below please find our responses to your comments. The revised manuscript is attached below wherein all changes we made are marked.

1.  The authors present a summary of their methodological developments in terrestrial photogrammetry for glacier monitoring. Unfortunately, most of the material in the manuscript has already been presented in a previous paper by the authors (Rosenau et al. 2013). Since the authors provide the reference to their previous study only on page 23 I was left with the impression to read through original material and spend the better part of a day on this review!

    The contribution that we submitted is meant to be an overview paper on recent photogrammetric methods and applications in glaciology, written by photogrammetrists and addressed to a broader readership. The contents published in the Rosenau-2013 paper play only a minor role therein, and there is no identical text or identical figures. Rosenau-2013 was actually never meant to be a methodological paper; the focus was clearly on the interpretation of the velocity fields and not on the algorithms to determine them. Of course there is a chapter covering the photogrammetric method as well, but it is only one page.

    From the following comments we got the impression that there is one main misunderstanding: Our main goal is to present the concept of our method in detail. In addition, some case studies are shown to show the potential of the methods; some of these case studies have been published before, but in a completely different context.

    We started time-lapse measurements and methodical developments in 2004 (Jakobshavn Isbrae, Greenland) and conducted further field campaigns in 2007 (Jakobshavn Isbrae, Greenland), 2009 (San Rafael Glacier, Chile), 2010 (Jakobshavn Isbrae, Equip Sermia, Store Qarajac, Greenland; Colonia Glacier, Chile), 2013 (Colonia Glacier, Chile), 2014 (Colonia Glacier, Grey-Glacier, Chile), 2015 (Kaskawulsh Glacier, Kanada). This was done in cooperation with various partners and with different investigation goals. The experiences gained through all of these campaigns were used to further develop our method. As we are photogrammetrists, our research focus is on methodical development and accuracy assessment/optimization. Rosenau-2013 thus is just one of several studies from which we gained experiences.

    We rephrased parts of the introduction to make the objective of our contribution become more clearly and to early address this previous paper.

    Detailed comments:

2.  The introduction provides a historical perspective on the use of terrestrial photogrammetry for glacier studies but I feel it leaves out some significant recent advances in the field of environmental monitoring with photogrammetric methods including the monitoring of glacier (e.g. Messerli and Grinsted 2015), landslides (e.g. Travelletti et al. 2013) or discharge monitoring (e.g. Stumpf et al. 2016). Some of those tools are readily available in the public domain and can be used for the proposed applications. It would therefore be helpful to explain briefly possible shortcomings of available methods and how the proposed method addresses those issues.

    Addtitional references:

    -   Messerli, A. and Grinsted, A.: Image GeoRectification And Feature Tracking toolbox: ImGRAFT, Geosci. Instrum. Method. Data Syst., 4, 23-34, 2015, doi:10.5194/gi-4-23-2015
    -   Travelletti, Julien, et al. "Correlation of multi-temporal ground-based optical images for landslide monitoring: Application, potential and limitations." ISPRS Journal of Photogrammetry and Remote Sensing 70 (2012): 39-55.
    -   Stumpf, André, et al. "Photogrammetric discharge monitoring of small tropical mountain rivers: A case study at Rivière des Pluies, Réunion Island." Water Resources Research 52.6 (2016): 4550-4570.

    We integrated the proposed references in the introduction.

3.  **Comment:** p.3, l.12: 'translation into object space' Maybe 'transformation' or 'mapping' would be more adequate than 'translation' here

    We changed this.

4.  p.5; l.9: "This can be achieved by establishing a local photogrammetric network (consisting of several convergent images taken from different positions as shown in Figure 2)" As I understand this, this authors suggest a single 3D reconstruction of the glacier surface for a single time steps as opposed to the time-lapse camera which will acquire time-series for the motion measurements? This implicitly comprises the hyphothesis that the glacier surface topography will not undergo large changes during the monitoring period. In particular for the observation of calving event this appears as a rather strong assumption which should be explained in greater detail.

    Correct, this is the limitation of a single camera system (monoscopic approach): a DEM is not available in the same temporal resolution as the time-lapse measurement series. This can only be provided by a synchronized multi camera setup. Section 2.1 discusses advantages and disadvantages of both approaches.

    In this contribution we are presenting the monoscopic approach and how we are going to deliver distance values for scaling. In contrast to other approaches (e.g. Box and Ahn, Messerli and Grinsted) we do not use an external DEM that needs to be registered with the time-lapse image (and will most likely be even less representative). We prefer an integrated solution to derive the DEM and its registration with the time-lapse image in one step. This offers high inner accuracy, good accordance with the cameras field of view and avoids registration errors. That's the statement of the section of which you picked the cited sentence from. Please note that the DTM is mainly used to determine scale factors to translate trajectory lengths from pixels to meters. Therefore, the requirements to the accuracy and resolution of the DTMs are not high. Our assessment has shown, that a DTM obtained from multiple images taken at the beginning and/or end of an image sequence is suitable to determine scales at an error <1%.

The special case calving event: The "rather strong assumption which should be explained in greater detail" is explained in the following sections (compare especially section 2.4.3). From the DEM we obtain a distance value for the starting point of a trajectory. We want to apply the monoscopic approach; therefore we need to make assumptions about the glacier motion. The basic assumption we make is, that glacier point is moving within a vertical plane which is oriented along the flow direction of the glacier. This allows us to distinguish between vertical and horizontal components of the motion vector as far as the glacier motion follows our model assumption. Let's assume the glacier is rising and falling close to the glacier front during a calving event while moving forward, these motions can be separated and measured. Problems would arise from motions across flow direction. The more the real motion deviates from the assumed motion model, the less accurately it can be split into its horizontal and vertical component. From the calving events that we observed we did not get the impression that there are significant across-flow motions that disturbed the velocity measurements.

5.  General comment on 2.2 Measurement Setup: While the section clearly describes several options for the measuring setup and states requirements (e.g. "should be replaced by differential GNSS measurements." "an elevated camera position is required", "it is necessary to have static points", "it is recommendable to use redundant information in the process of geo-referencing"). While I agree with those considerations it is still not clear for me which (similar or different setups and strategies were used in the study. I would encourage the authors to provide more explicit information on the actually realized setups.

    It's the general methodology that we want to present in detail with this contribution, not a certain study. All strategies we suggest were realized and tested in various field campaigns (see answer to comment 1). Each measurement area is unique and may put individual restrictions to measurement setup and data processing. We wanted to provide a method that can handle most of these situations, and we thus integrated accordant options in our software. So for each individual measurement case/ application, the user must decide, which of the provided strategies is the most appropriate. Say in this section the one who developed the method tries to give recommendations on which option best fits which situation.

6.  p. 8: "Thus, the first strategy might be applied when tracking signalised points" Similar to my previous comment, please avoid hedging, and state what has been actually done.

    Ok agree, this sentence sounds too vague. Changed "might" into "has to be".

7.  p. 12, l.7: 'higher than the single standard deviation' The standard deviation of the gray values within the search patch? This automatically assign several pixels as shadow pixels even if not shadows are present at all. It seems to me that on a glacier this might mask many areas with important texture and could leave you only with the areas that are difficult to match? Please comment.

    That's why it is an iterative process. After the first iteration step the state is as you describe it above: Some shadow motion pixels are removed but also some non-shadow motion pixels. The removal of the shadow motion pixel now is what makes the following iteration converge to a better solution. This means more shadow motion pixel but less non shadow motion pixel (less "important texture") are removed during the subsequent iteration step. In order to find the best compromise to exclude as much shadow motion pixel but at the same time keep enough of the glacier texture to allow for a successful next matching we look at the difference image of the master patch defined in image 1 and its corresponding patch detected in image 2. High differences indicate shadow shifts on the one hand but also important glacier texture. Now we have to find an appropriate threshold that removes disturbing shadow influenced pixel in order to guarantee a better solution in the next matching step but keep enough good glacier texture for the matching as well. When looking at the histogram of the difference image the difference values are approximately normally distributed in most cases. The shadow influenced pixels show the highest grey value differences and accumulate at the upper and lower margin of the histogram. Thus, using the single standard deviation of the difference values as a threshold ensures that no more than ca. 30% of the pixels of a patch are excluded and enough texture remains in the patch. Our procedure also consists of a solution for non-normally distributed cases but that would probably lead to far for the paper.

    We rephrased our explanations in order to make the algorithm become better understandable.

8.  p.13, l.2: "error influence on the motion vectors" This sounds a bit awkward. I suggest something like 'error source that might impact'...

    Agree, we changed this as you suggested.

9.  p.14, l.9: "two image shift parameters and the rotation parameter" If you are only considering those two parameters you will estimate a rigid transformation and won't need to invoke an affine transformation. Please clarify.

    You are right; as it is described in the paper we should have called it a rigid transformation.
    However, in our procedure it is implemented as an affine transform with the option to also estimate scale parameters. We used them to test whether refraction affects the measurement of the fix targets or as an indicator for changes of the interior orientation of the time lapse camera. Since these issues are not part of this paper we changed the term into "rigid transformation".

10. p.16, l.10: "Going a step further and calculating the sparse cloud while providing the measured camera positions to PhotoScan, the thus determined 3D coordinates of object points can also be exported and used as approximate values for the bundle block adjustment." I do not understand the purpose of this step. Errors in the internal and external calibration parameters plus matching errors will propagate into the sparse point cloud. Running another bundle adjustment with the resulting 3D points will hence certainly suffer from those errors? Please clarify.

    During a bundle block adjustment a non linear equation system needs to be solved. The process of linearising the equations requires approximate values for each unknown parameter. Running our own bundle in order to calculate 3D glacier point's means we need an appropriate start value for each coordinate that will converge to an optimal final value during the iterative adjustment procedure. We just take advantage of the sparse point cloud of PhotoScan as a source of approximate start values for our own bundle (which – unlike PhotoScan – does not have a strategy for automatic approximate value generation). Therefore possible errors of the sparse cloud don't matter because these values are all estimated anew, as well as the external orientation parameters of the camera.

Internal calibration parameters of the used cameras were always externally estimated with high accuracy beforehand. For this purpose we used an adequate calibration field and photogrammetric calibration software. In the following process they don't need to be estimated anew and are thus set as fix values (in PhotoScan as well as in our own bundle).

11. Similar too my previous comments p.16 comprises a lot of ambiguity. Some example: "we predominantly used structure from motion (SFM) tools (such as Agisoft PhotoScan)" Which other SFM tools where used and for what?

This contribution (which is meant to be a conceptual methodological paper) is based on experiences gained during a variety of different studies between 2004 and 2015:

With "Bundler", the first structure from motion library was first released in 2008. PhotoScan started in 2010. After that the development in this field rapidly increased. This means for the processing of all measurement campaigns before 2011 we solely had to rely on our own bundle. Then, we started to use "Bundler" for the measurements of corresponding image points and finally started to integrate the functionalities of PhotoScan into our work flow. Thus we "predominantly used structure from motion (SFM) tools (such as Agisoft PhotoScan) in combination with our own photogrammetric bundle".

There is still the option to solely use our own photogrammetric bundle, which is completely flexible and adaptable to different geo-referencing concepts (e.g. using height control points, introducing scale conditions, calculating the individual error for each estimated parameter,…), but requires a deeper photogrammetric understanding from the user. PhotoScan as a commercial software on the other hand is easy to handle, fast and optimized. To use the advantages of both, we had to define an appropriate interface between them. It depends on the individual geo-referencing setup of a measurement scene, up to which state of the workflow we can use PhotoScan or need to switch to our own software.

12. "It can be adapted to different types of control points as well as different sets of camera calibration parameters, scale conditions can be defined, and it provides the possibility to define each variable as fix or parameter to be estimated". Which type of cameras and camera calibration models where used? How did you adapt the tool and the free parameters and based on which criteria?

We simply wanted to state that an open photogrammetric bundle has advantages in comparison with the PhotoScan bundle. The latter cannot be used as flexible as an open bundle, especially regarding the issues mentioned in the cited sentence. An open photogrammetric bundle just offers more options. Thus, we also have more options when deciding for a certain measurement setup in the field.

To answer your special questions:

- type of cameras:    We used central perspective cameras during our campaigns, mainly mirror reflex cameras. However our in-house bundle contains of different fisheye-camera models, a panorama camera model and a laser scanner model as well. So if we e.g. would insist to use a GoPro as time-lapse camera we could easily switch to a fitting fisheye model.

- calibration models:    We mostly used the calibration model as described in Luhman (2006), because this is widely used in photogrammetry and also implemented in the external calibration software we used (AICON 3D Studio). (*Luhmann, T; Robson, S; Kyle, S; Harley, I.: Close Range Photogrammetry: Principles, Methods and Applications. [Book]. (1st ed. ed.). Whittles: UK, 2006*). Computer vision calibration software might use other sets of calibration functions. PhotoScan also uses an individual calibration model that even varies between different program versions. If we pre-calibrate cameras and thus have to process our data with different software we need to overcome such inconsistencies. In this respect, the advantage of an open bundle is that each type of calibration model can be integrated as far as it is openly documented.

- free parameters:    If we use height control points we have to fix the Z-coordinate of a control point but let X and Y free to be estimated.

However, we think it would be a bit out of scope to explain all of these special aspects in detail. The paper already consists of 30 manuscript pages, and we should probably not extend it to a textbook.

13. "Since many SFM tools are rather optimized for fast processing and 3D-visualization than for accurate measurement purposes, some limitations may have to be taken into account, when applying them for measurement tasks."Which tools are you talking about? Can you back this with numbers or previous studies?

Most SFM-tools like Bundler, PhotoScan, VisualSFM,… originate from computer vision. In computer vision the focus usually is on optimizing algorithms for fast processing and 3D-visualization whereas the focus of photogrammetric approaches is on optimizing algorithms for high accuracy. Bundler e.g. does not provide an integrated geo-referencing, PhotoScan turns more and more into a photogrammetric software, but it still doesn't provide information about the individual error of each estimated parameter and doesn't provide flexible parameterizations such as the use of height control points.

We added some references which back up this statement:

Eltner, A., Schneider, D.: Analysis of Different Methods for 3D Reconstruction of Natural Surfaces from Parallel-Axes UAV Images. In: The Photogrammetric Record, 30(151), pp. 279–299, 2015

James, M.R., Robson, S., d'Oleire-Oltmanns, S., Niethammer, U.: Optimising UAV topographic surveys processed with structure-from-motion: ground control quality, quantity and bundle adjustment, In: Geomorphology, 280, pp. 51–66, 2017

14. "However, when not using a reduced measurement setup it is also possible to determine object coordinates and camera orientation parameters solely using PhotoScan." What is a reduced measurement setup? Is this an option or something you have done in this study?

   In section 2.2 we introduced several possible geo-referencing setups which require different measurement equipment and effort. In the field or during expedition planning the choice has to be made which of these geo-referencing setups fits best depending on:
   - the topography of the individual measurement scene
   - how much measurement equipment we want to carry in the field
   - how much time we have to conduct the geo-referencing measurements
   - or even which kind of processing software we want to use

   The chosen geo-referencing setup than decides up to which state of the geo-referencing workflow we can use PhotoScan or need to switch to our own software. To use PhotoScan as much as possible (which might be something non-photogrammetrists would prefer) we need to choose the most extensive one from the proposed geo-referencing setups (a "non-reduced" measurement setup) because full 3D control points are required.

   For practical reasons we used "reduced measurement setups" during most of our studies because they save time and money. For their processing a flexible photogrammetric bundle is required. This can be combined with certain functionalities of PhotoScan to obtain the input data for the bundle (image coordinates, approximate values). In contrast, the full geo-referencing setup will work in any case and has the advantage that the bundle adjustment can completely done with PhotoScan but the measurement in the field is time consuming and equipment-intense.

   We restructured and rephrased some text in section 2.2 and 2.4 to make this become more clearly. We numbered the proposed geo-referencing options to more clearly refer to them in section 2.4.

15. "Thus, it is recommendable to measure the cameras positions in the field and to precalibrate the cameras..." I agree, but it is again not clear if that is a recommendation or something you implemented in this study. I there are more examples like this.

   This does not refer to a certain study. The pre-calibration is something we did during all of our measurement campaigns. From our photogrammetric point of view we recommend this. The sentence you cited above continues with the reasons why.

16. p.17, l.13: "In case of using PhotoScan, the exterior camera orientation parameters and a depth map can be exported for the time-lapse image." Above you explained that you use you own in-house bundle adjustment and in combination with pre-calibration of the internal parameters. Why would you then export these parameters from Photoscan?

   If we choose the most extensive one from the proposed geo-referencing setups (a "non-reduced" measurement setup) and measure 3D control points instead of height control points, we have the opportunity to use the bundle of PhotoScan and don't need to switch to our own bundle (compare answer to comment 14).

   If we pre-calibrated the cameras the interior orientation parameters are known and can be introduced as fix values into the bundle (in this case the PhotoScan bundle) as well as the 3D control points and the camera positions. The results we need to obtain from the PhotoScan bundle are the exterior orientation parameter of the time-lapse camera. From the dense matching step we need to obtain the depth map of the measurement object (a depth value for each pixel of the time-lapse image).

   This is what defines the interface between PhotoScan and our software in case of using a "non-reduced" measurement setup: Exporting exterior camera orientation parameters and a depth map from PhotoScan and importing them, together with the pre-calibrated interior parameters, into our software. Here they serve to scale the trajectories and define their position in a world coordinate system as explained in section 2.4.3.

17. p.18, l. 12: "The glacier flow direction can e.g. be obtained via flow-line patterns that are visible in satellite orthophotos. Does this imply that the glacier flow direction is constant across the scene (as shown in Fig 12A)? This would be a rather strong assumption for observations in areas where the glacier flow is bending with the topography. Please clarify and state possible limitations that may arise from this assumption.

   In section 2.1 we compared advantages and disadvantages of using a single time lapse camera and a multi camera time-lapse system. A full 3D motion acquisition (via multi-temporal laser scanning or multi-camera time-lapse) would directly offer 3D-motion vectors. Since we are using single camera systems, we have to make assumptions about the motion direction.

   The simplest assumption is orthogonality between glacier flow and camera viewing direction. The next step of improvement would be to introduce an overall mean flow direction of the measured glacier part (as we propose as a local approximation in the contribution). From our own experience, this will be sufficient for most of the tasks, as time lapse observations usually cover small parts of a glacier.

   Of cause there may be single cases (strong bending glacier + large observation section + wide angle lens + high accuracy demands), where we would need to go even further. In these cases the accuracy can be increased by introducing flow line polygons (again digitized from Satellite imagery) which provide an individual local flow direction for smaller parts of the measurement area.

18. p.19, l. 16: "For a sample trajectory from an image sequence measurement at Jakobshavn Isbræ in May 2010" I feel it would be helpful to first introduce the study sites with their respective measurement setups before providing results on the accuracy. At this point it is rather difficult for the reader to understand if the error budget estimation is representative for all sites.

   To complete the presentation of the method we wanted to name the main error influences of the method and exemplarily show the magnitude of accuracy that can be achieved.

   We added some more specifications of our measurement example.

19. p.20, l.20: "These individual errors were propagated into a mean total error of 9.2 cm "It might be useful to provide the formula for the error propagation. Why did you not also assess the measured velocities against independent in-situ measurements?

It would be out of scope of the paper to present the whole error analysis for the method. The general mathematical principals of geodetic error estimation theory can be found in textbooks (e.g. Niemeier, 2002 or Taylor, 1997). These references are given in the paper. An individual formula to calculate the error does not exist because the whole method cannot be described with a single formula. The error propagation is inherently incorporated in all photogrammetric routines used here, including bundle adjustment and least squares image matching.

Of cause we also compared the time-lapse measurements with independent measurements, including terrestrial laser scanner measurements, total station measurements, satellite based measurements. However, since these methods cannot be assumed to have a higher accuracy potential than the time lapse method, they cannot be used as a reference for absolute error calculation. This comparison only proofs that the results are plausible and in accordance with other methods. To derive an individual error value for each measurement value, we still have to rely on a thorough inner accuracy analysis of the method.

We are currently planning to write another paper that will cover the topic of error analysis for the presented method. The subject is too extensive to be described in detail here.

20. General comment related to section 3 Assessment of glacier motion fields: four case studies. I feel that a lot of important information regarding the case studies is missing including maps of the measurements setup, the type of deployed cameras, the duration and frequency of the acquisitions of monocopic images as well as stereo views. Introducing the particularities of the study sites might also help the readers to understand which specific choices were made following the rather generic description in section 2.2. Another interesting aspect that might deserve some further considerations is the operation of the camera systems in the those rather harsh environmental conditions (e.g. power supply, data storage and submission, etc.).

This section was never meant to present the results of these applications as new results. As the introduction of the section tells we wanted to: "…present examples for different glaciology applications of the method to show its potential". It's a discussion about the suitability and possible areas of application of the method. For more details, references are given to application-oriented publications where more information can be found on the individual studies.

We admit that the headline of the section is misleading here. Subscribing the section with the Term "discussion" as referee 2 suggests will hopefully prevent from further misunderstandings.

21. 3.1 Horizontal glacier motion. Is the glacier motion in the presented case studies purely horizontal? If not your measurements may also comprise a vertical component. Please clarify.

We always do measure a horizontal and a vertical component of glacier motion. In section 3.1 we wanted to discuss examples of applications that focus on the horizontal component of the derived motion vector.

22. 3.1.1 Glacier motion velocity fields
Considering all the effort for setting up a time-lapse system it is more than unfortunate that you only present the aggregated means. Surely you derived some interesting timeseries that you could present? What can we learn from those time-series regarding the process of glacier flow? Section 3.1.2 is much more mature in this regard.

See answer to comment 20.

23. p. 23, l 14: "velocity increase decreased with increasing" There is a lot of increase and decrease here. Please reformulate.

We changed this.

24. Figure 16: I would be helpful to provide a scale for the upper figure (e.g. distance between to arbitrary points on the stable terrain). Please also provide the details on the smoothing filter (e.g. rolling mean with a window size of x?) used.

As Referee 2 suggested we now changed the sections headline into "Discussion", removed the figures and shortened it.

25. 1.1 Vertical glacier motion Something went wrong with the section numbering here.

Thank you, we corrected this.

26. 1.1.1 Grounding line determination
There are several issues in this section. The caption of Figure 17 refers to only two of the 3 subfigures and raises the impression that all the results presented originate from other studies.

Our objective was not to present the results of these applications as new results. We rather wanted to show the potential and value of the method for different types of glaciological applications. Thereby the results of our previous studies should serve as examples and proof of the methods suitability for diverse tasks with different demands. Thus, for each example that has been already published we added the accordant reference.

As Referee 2 suggested we now changed the sections headline into "Discussion", removed the figures and shortened it.

27. The letters A-C used to number the subfigures are not used at all. In Fig 17 A it is not clear which measurement is from which source. The y-axis gives the impression that both curves show the tidal range while one of them shows (I assume) the vertical component of the glacier motion. "Figure 17 (A) shows the vertical motion component of a single trajectory compared to tidal

height measured by a pressure gauge." Neither from the text nor in the figure it is clear which measurement is which. "In particular, the scale factor in vertical direction" The paper is not really self-contained here. Please explain how the scale factor is computed.

The figure has been removed following the suggestion of Referee 2 to shorten the section.

28. "Multiple time lapse measurements at Jakobshavn Isbræ in 2004, 2007 and 2010 even allowed for the documentation of the migration of the glaciers grounding line (Figure 17, C) (Rosenau et. al., 2013)." I finally had a look at the study by Rosenau et al. 2013 and came to realize that an important part of the presented material has already been published in this previous paper by the authors. This includes the measurements on the Jakobshavn Isbrae glacier, the corrections for camera movement, the projection into object-space, estimation of the error budget, the delineation of the grounding line, and the documentation of the calving event. I stop the review here.

We would really like to ask you to continue!

We never claimed that "the measurements on the Jakobshavn Isbrae glacier, the delineation of the grounding line, and the documentation of the calving event" are new results. We did not hide that they are already published but added the references. In section 3 we wanted to show and discuss the potential and value of the method for different types of glaciological applications. Thereby the results of our previous studies should serve as examples and proof of the methods suitability for diverse tasks with different demands. The same holds for the results of other pilot studies shown in this paper.

Of course we mention camera motion and transformation into object space in both of the papers. These are essential parts of each monoscopic time-lapse approach. In Rosenau-2013 only a short method overview (1 page) is given as add-on to better understand where the measurements, which are analyzed and interpreted in the paper, come from (the reviewers especially requested it that time). The part of presenting the individual measurements at Jakobshavn Isbrae and their analysis and interpretation was the intention and main focus of this paper, not a detailed presentation of our photogrammetric time-lapse method.

Our submitted contribution has about 20 manuscript pages of pure and detailed method description + 5 pages of discussion of the methods potential and suitability supported by application examples. And the review shows that there are still several open questions concerning the method. Isn't this in a certain contradiction to the statement that everything has already been published?

[revised manuscript text omitted]

---

## Author Comment (AC2) · 31 Aug 2017

**Answers to Anonymous Referee #2**

We want to thank the referees for their comments and suggestions and for the time they spend reviewing our manuscript. Below please find our responses to your comments. The revised manuscript is attached below wherein all changes we made are marked.

1. I have read the paper with interest, and I think this is a potential interesting contribution for the special issue as it matches very well with the topic. My main concern about the paper is the very large similarities with the paper already published by Roseneau et al. 2013. Indeed this paper is not cited in the introduction when I think that it should be clearly stated that this paper exists and which is the exact contribution and adding of this new paper.

   The contribution that we submitted is meant to be an overview paper on recent photogrammetric methods and applications in glaciology, written by photogrammetrists and addressed to a broader readership. The contents published in the Rosenau-2013 paper play only a minor role therein, and there is no identical text or identical figures. Rosenau-2013 was actually never meant to be a methodological paper, the focus was clearly on the interpretation of the velocity fields and not on the algorithms to determine them. Of course there is a chapter covering the photogrammetric method as well, but it is only one page.

   We rephrased parts of the introduction to make the objective of our contribution become more clearly and to early address this previous paper.

2. It would also allow to prepare a revised version more focussed and shorter, as in its current state it seems to me that it results a little bit sparse and too long (18 figures is probably excessive for a paper). As a suggestion, I think that the paper will be more interesting for readers focussing in the detailed presentation of the methodology and validation that you do that exceeds the one offered in Roseneau et al.2013. Much of the sections dealing with applications and results could be removed and synthetized in a discussion section. I would avoid to exceed 10-12 figures in total.

   Our objective was not to present the results of these applications as new results. We rather wanted to show the potential and value of the method for different types of glaciological applications. Thereby the results of our previous studies should serve as examples and proof of the methods suitability for diverse tasks with different demands.

   We followed your suggestion and transformed the case study section into a shorter discussion section.

3. I think it would be useful to add a figure showing a scheme with the complete workflow applied to the figures in a sequential way.

   We added the requested figure in section 2.1.

   Minor comments:
4. In general the methodology is presented in a clear way and it is possible to be understood by the majority of readers, but some sentences could be simplified to facilitatethe lecture for people not very familiarized with this technique. An example is. "We limit ourselves to monoscopic image motion capture and processing delivering two-dimensional velocity field information here, as the 15 glaciology phenomena observed in the practical experiments do not show significant across-track motion and can thus be well described in 2D."; or to explain better some concepts as "decorrelation" used in page 2 line 29.

   We tried to rephrase these sentences.

5. It is necessary to read more than half of the paper to realize that you are using Photoscan and a library developed for yourselves for the analyses. I would mention this earlier in the paper and I would indicate if the library is open for all users.

   We now mention it already in section 2.1.

6. P7 line 19: First, not fist.

   Thank you, this has been corrected.

7. P6, line 23. "The cameras camera position...."

   Thank you, this has been corrected.

8. How is corrected the distortion of the lens? Using the info provided by the producers or using some tool as the one provided by Photoscan?

   For camera calibration we used the photogrammetric software "Aicon 3D Studio" and coded marks that can be measured with sub-pixel accuracy. We usually try to install a calibration area in the field and calibrate the cameras shortly before taking measurement images or installing them as time-lapse cameras.

   We added this information in the text.

9. If I understood well, all the analyses were made with only one camera. In Figure 2 the inclusion of more photogrammetric cameras may lead to confusion.

The figure mainly illustrates the camera network that is necessary for scaling and georeferencing of the time-lapse measurements. Since the time-lapse camera has to be part of this network to obtain its orientation and position in relation to the 3D surface geometry of the glacier we integrated all camera positions in the scheme. With the image we want to show that the camera views need to have some convergence that, they should cover the area observed by the time-lapse camera and that control points should be visible in all of these images.

We would therefore prefer not to change the scheme and added some explanations in image caption and text to prevent from the confusion you mentioned.

10. Please check well the references-citations as there are several mistakes.

Thank you for the hint, we checked and corrected remaining formatting errors.

11. Indeed the reference of the paper Roseneau et al. 2013 is wrong.

We added the missing line break in the reference list. But in the reference itself we couldn't find any mistakes. The name of our colleague really is Rosenau.

[revised manuscript text omitted]

---

## Editor Decision (ED1)

**LETTER TO AUTHORS**
* * *
**Journal:** Earth Surface Dynamics

**Special Issue:** 4-D reconstruction of earth surface processes: multi-temporal and multi-spatial high resolution topography

**Paper Title:** *Determination of high resolution spatio-temporal glacier motion fields from time-lapse sequences*

**Manuscript ID:** esurf-2017-33

**Date:** 06- Oct - 2017
* * *
Dear authors,

Thank you very much for submitting a reviewed version of your manuscript esurf-2017-33. First of all, I would like to congratulate for your intense work answering all comments from the referees and reviewing the first version of the manuscript accommodating the majority of their suggestions.

I have received and reviewed all reports provided by the referees. The two referees that reviewed the first version of the manuscript accepted to review the new version of the paper. Additionally, the paper has been reviewed by a third referee to have an independent view of the manuscript (i.e. a view of a specialist on the topic that has not seen the first version of the paper). The two referees that reviewed the first version of the manuscript agreed in that the paper has gain clarity. Although referee #1 commented that 'the paper still suffers somewhat from the difficulty to condense more than one decade of research in one coherent and self-contained manuscript', at the end, both referees have accepted the paper as is.

In terms of the new referee (#3), although the referee considers that the paper has an excellent scientific quality, the final decision of this referee has been considered for publication after major revisions. After evaluating the three reviews my final decision is the paper can be accepted after minor revisions.

These minor revisions are based on Referee's #3 recommendations. I consider these recommendations/suggestions can improve the scientific significance and quality of the work. In the following sections I have provided a summary of the strengths and few suggestions the authors should address before the paper is finally accepted for publication. All of these are based in the positive suggestions provided by referee #3 (note that some of the referee's sentences are literally used/provided in the following sections).

Strengths:

- Referee #3 considers that the authors have an excellent track record in this area, in terms of applying rigorous approaches in demanding conditions, and in developing novel algorithms capable of coping with real-world issues such as shadowing.

- Referee #3 indicates an underlying strong support for a review-style paper within ESurf (particularly for this special issue) to highlight the authors' work to a highly relevant end-user community. The referee pointed out that the goal of the paper is not to deliver new glaciological understanding, but to provide an overview of the methods and their applications and could be of value to a wide range of ESurf readers. This was also answered and discussed in the answers you provided after the first review of the manuscript, making clear the objective/goal of the paper, which I confirm fits very well to this Special Issue and it can be of wide appeal to the readership.

Suggestions:

- Increased conciseness in places and providing insightful, quantitative comparisons with other published work would substantially strengthen the paper and increase its impact. As a suggestion, that I also share, referee #3 proposes: "the introduction provides a number of other scenarios where time-lapse photography has been used, but this is done in a somewhat descriptive list-like manner (paragraph 3, Page 2). A more critical analysis of these preceding works (and this maybe only really needs to be of the glaciological ones) would help put the authors' contributions in better context. For example, the basic details of the papers could be collated in a table - e.g. to list process being monitored, stereo/mono, image interval/duration, feature or area matching and algorithm, approximate accuracies achieved etc., - then the key advantages/limitations of these works discussed in a way that highlights how the authors' advances addresses particular limitations. Image registration/camera stability is an obvious area in which previous authors have done work; a clear recognition of this would enable the quality/flexibility of the approaches covered in the review to be discussed/compared rather than just described. How much better are the authors' camera registrations than those achieved by others? What do we need to do to improve things?"

- Referee #3 suggests renaming the discussion to case studies. I'm aware the authors already changed the name of this section to discussion following a suggestion of one of the referees in the first review of the manuscript. Additionally, in page 25, first paragraph, it's already stated 'The discussion is based on the results of previous case studies on determining glacier motion data with the method presented here'. Therefore, in my opinion, I do not thing renaming the section is an essential change. Having said that, one suggestion I do thing would improve the manuscript is trying to be more critical in the conclusions, as pointed out by the referee, highlighting the advantages or advances of the authors' specific approach in comparison with similar work (as the referee commented, "Can the authors pick out the key contributions from their work (e.g. dealing with shadows) and focus on these to give a more concise and inspiring summary of their advances?").

- Finally, the referee also suggests if the software (stated in page 28, acknowledgments) is already available and if there is any possibility to be linked directly in this manuscript. As the referee indicated, "this would be an excellent way of providing

something substantially new within the work; I encourage the authors to try and do this".

Minor queries/suggestions:

Additionally, referee #3 has provided some minor queries/suggestions that in my opinion also deserve consideration:

- P2; paragraph 3: volcanological applications could also be relevant, e.g. .papers by Walter's group on domes: 10.1111/j.1365-246X.2011.05051.x; 10.1002/2016JB013045; 10.1002/jgrb.50066; or USGS: doi.org/10.1016/j.epsl.2009.06.034; or other on lavas: 10.1007/s00445-011-0513-9; 10.1016/j.isprsjprs.2014.08.011

- P2; Line 32: Should this be James et al (2016)? This work dealt with very difficult imagery, for which fully automated tracking was not possible – an interesting comparison for discussion elsewhere in the paper?

- P5; L5: Ensure that photogrammetry terminology is either avoided or explained carefully at first use. Here, maybe rephrase/explain 'inner accuracy'?

- P6; L11: Replace 'avoid' with 'reduce'.

- P8; L9: The phrasing could indicate that the algorithms cited are specifically for glacier point tracking, whereas the references are for general image registration. Rephrasing would clarify, e.g. 'A wide range of algorithms are available for point tracking in image sequences (e.g….'

- P10; L5/19: 'imported' → 'important'

- P18; L14: So which variants from P 7 can be employed with PhotoScan?

- P18; L19: 'to warrant up-to-dateness' → 'To ensure continued validity'?

Finally, I would like to acknowledge all the work the authors have done that clearly improved the first version of the manuscript, and to thank all the positive comments and suggestions provided by the three referees that helped in improving the scientific significance and quality of the manuscript. Having said that, before to finally accept the paper for publication, I would like the authors consider the suggestions that referee #3 provided that clearly will help in improving the strength of this manuscript.

All the best
Damià Vericat
(acting as associated editor of the Special Issue *4-D reconstruction of earth surface processes: multi-temporal and multi-spatial high resolution topography*)

---

## Author Response (AR2)

**Answers to "Letter to authors"**

Thank you for the review and summary of the referees reports. We also want to thank referee#3 for reviewing our manuscript and his comments. Below please find our responses to your suggestions. The revised manuscript is attached below wherein all changes we made are marked.

Suggestions:

1. Increased conciseness in places and providing insightful, quantitative comparisons with other published work would substantially strengthen the paper and increase its impact. As a suggestion, that I also share, referee #3 proposes: "the introduction provides a number of other scenarios where time-lapse photography has been used, but this is done in a somewhat descriptive list-like manner (paragraph 3, Page 2). A more critical analysis of these preceding works (and this maybe only really needs to be of the glaciological ones) would help put the authors' contributions in better context. For example, the basic details of the papers could be collated in a table - e.g. to list process being monitored, stereo/mono, image interval/duration, feature or area matching and algorithm, approximate accuracies achieved etc., - then the key advantages/limitations of these works discussed in a way that highlights how the authors' advances addresses particular limitations. Image registration/camera stability is an obvious area in which previous authors have done work; a clear recognition of this would enable the quality/flexibility of the approaches covered in the review to be discussed/compared rather than just described.

   We fully agree with referee #3 that a review paper on the subject has not yet been written and that it would be of high interest for scientists in this field. We ourselves would highly appreciate such a paper. But the paper we submitted was not primarily meant to be a review paper, it was rather written as a methodical paper introducing our approach of terrestrial time-lapse measurement. The manuscript has undergone an intensive review already and the first two referees now agreed to it.

   At this stage, we do not want to completely change the focus of our paper by turning it into a method review. We understand that the proposed table would be a compromise to somehow refer to referee#3's main comment.
   But each measurement application is very different: There are reasons to use high resolution or low resolution cameras, wide angle lenses or narrow angle lenses, camera housing with optical glass, simple window glass,...  Beside these technical specifications of the measurement setup, the measurement results are influenced by the objects size and velocity, the terrain and the weather conditions of the measurement set. The scientific question defines the recording time interval, the absolute time of observation and the accuracy requirements. It seems impossible to condense all this into a table, and as a consequence such a table would also earn criticism. Therefore we would prefer to leave it with the "descriptive list-like manner", in order to give the reader an overview who else has been working on the subject, but without claiming completeness.

   How much better are the authors' camera registrations than those achieved by others?

   We stated that if we calculate the DEM and the camera orientation together in one adjustment, this will result in better accuracies for the scaled trajectories than for the registration of an image sequence to an external DEM (usually done via spatial resection on the basis of corresponding points, that need to be defined in the image and on the DEM). This is a qualitative assessment that can be made by considering the fact that in the case of a bundle adjustment that integrates the time-lapse camera each object point of the DEM contributes to the image registration whereas in the case of the external DEM only the

measured corresponding points will do so. Furthermore, the errors of an external DEM will have an influence on the image registration and thus propagate into the measured trajectories. A quantitative comparison depends very much on the actual application and can only be made if the different methods are applied to one and the same measurement object.

What do we need to do to improve things?"

More research of course … ;-)
This is a question, which also can hardly be answered in general because it depends on the individual conditions of a specific measurement task as described above. Reliable energy-saving high-resolution camera sets would help a lot in many applications – but this is not primarily a scientific problem. On the algorithmic side, future research might for instance look deeper into feature-based image matching techniques.

2. Referee #3 suggests renaming the discussion to case studies. I'm aware the authors already changed the name of this section to discussion following a suggestion of one of the referees in the first review of the manuscript. Additionally, in page 25, first paragraph, it's already stated 'The discussion is based on the results of previous case studies on determining glacier motion data with the method presented here'. Therefore, in my opinion, I do not thing renaming the section is an essential change. Having said that, one suggestion I do thing would improve the manuscript is trying to be more critical in the conclusions, as pointed out by the referee, highlighting the advantages or advances of the authors' specific approach in comparison with similar work (as the referee commented, "Can the authors pick out the key contributions from their work (e.g. dealing with shadows) and focus on these to give a more concise and inspiring summary of their advances?").

We shortened the conclusion and rephrased parts to highlight our key contributions. (pp. 27-28)

3. Finally, the referee also suggests if the software (stated in page 28, acknowledgments) is already available and if there is any possibility to be linked directly in this manuscript. As the referee indicated, "this would be an excellent way of providing something substantially new within the work; I encourage the authors to try and do this".

We fully agree. The requested link is now provided in the paper on page 28. The software will presumably be available on the linked webpage by the end of this year.

Minor queries/suggestions:

4. P2; paragraph 3: volcanological applications could also be relevant, e.g. .papers by Walter's group on domes: 10.1111/j.1365-246X.2011.05051.x; 10.1002/2016JB013045; 10.1002/jgrb.50066; or USGS: doi.org/10.1016/j.epsl.2009.06.034; or other on lavas: 10.1007/s00445-011-0513-9; 10.1016/j.isprsjprs.2014.08.011

Thank you! We added the aspect of volcanological applications and two of the proposed references (p.2, l.17).

5. P2; Line 32: Should this be James et al (2016)? This work dealt with very difficult imagery, for which fully automated tracking was not possible – an interesting comparison for discussion elsewhere in the paper?

This was a mistake; we changed the year to 2016

6. P5; L5: Ensure that photogrammetry terminology is either avoided or explained carefully at first use. Here, maybe rephrase/explain 'inner accuracy'?

   We removed 'inner' on page 5 because the term' inner accuracy' is explained on page 17, where it occurs for the first time now.

7. P6; L11: Replace 'avoid' with 'reduce'.

   We changed this

8. P8; L9: The phrasing could indicate that the algorithms cited are specifically for glacier point tracking, whereas the references are for general image registration. Rephrasing would clarify, e.g. 'A wide range of algorithms are available for point tracking in image sequences (e.g....'
   That's true, we changed this

9. P10; L5/19: 'imported' → 'important'

   We changed this.

10. P18; L14: So which variants from P 7 can be employed with PhotoScan?

    We added the accordant information.

11. P18; L19: 'to warrant up-to-dateness' → 'To ensure continued validity'?

    We changed this.

[revised manuscript text omitted]

---

## Editor Decision (ED2)

**LETTER TO AUTHORS**

**Journal:** Earth Surface Dynamics

**Special Issue:** 4-D reconstruction of earth surface processes: multi-temporal and multi-spatial high resolution topography

**Paper Title:** *Determination of high resolution spatio-temporal glacier motion fields from time-lapse sequences*

**Manuscript ID:** esurf-2017-33

**Date:** 11- Nov - 2017

Dear authors,

Thank you very much for submitting a new reviewed version of your manuscript esurf-2017-33. First of all, I would like to congratulate for your intense work answering all comments from the referees.

I would like to inform that the first version of the paper was reviewed by two referees. You did review the first version of the manuscript accommodating the majority of their suggestions. After receiving the reviewed version of the manuscript, I sent the manuscript to the same two referees that reviewed the first version, and to an additional referee. The two referees that reviewed the first version of the manuscript agreed in that the paper gained clarity and, at the end, both referees accepted the paper as is. The third referee provided some suggestions and some minor comments that needed to be addressed before to accept the paper for publication; although the referee considered that the paper had an excellent scientific quality.

I have received now your point-by-point answer to all comments referee #3 provided. After looking to all your answers, reviewing how these were addressed in the new version of the manuscript, and considering the decisions already provided by referee #1 and #2, I'm glad to inform the paper has been finally accepted for publication.

Finally, I would like to acknowledge all the work the authors have done that clearly improved the different versions of the manuscript, and to thank all the constructive feedback and positive comments and suggestions provided by the three referees that helped in improving the scientific significance and quality of the manuscript.

All the best
Damià Vericat
(acting as associated editor of the Special Issue *4-D reconstruction of earth surface processes: multi-temporal and multi-spatial high resolution topography*)